# Near-infrared-II photoacoustic imaging and photo-triggered synergistic treatment of thrombosis via fibrin-specific homopolymer nanoparticles

Jianwen Song[1], Xiaoying Kang[1], Lu Wang[2], Dan Ding [1], Deling Kong [1] ✉, Wen Li [2] ✉ & Ji Qi [1] ✉

The formation of an occlusive thrombus in the blood vessel is the main culprit for numerous life-threatening cardiovascular diseases that represent the leading cause of morbidity and mortality worldwide. Herein, we develop a polymer nanoplatform that integrates long-wavelength second near-infrared (NIR-II) photoacoustic imaging-based thrombosis detection and antithrombotic activity. We design and synthesize a semiconducting homopolymer with strong absorption in the NIR-II region and molecular motion that boosts photothermal conversion and photoacoustic signal. We dope the homopolymer with a thermosensitive nitric oxide donor to formulate a nanoplatform, on which a fibrin-specific ligand is functionalized to ensure selective thrombus targeting. We show that with strong NIR-II light harvesting capability, bright photoacoustic signal and active thrombus accumulation ability, the NIR-II photoacoustic nanoprobes are able to sensitively and selectively delineate thrombi. We find that the nanoplatform also displays rapid and efficient blood clot removal activity with nearly complete blood flow restoration in both carotid thrombosis models and low extremity arterial thrombosis models under NIR-II light trigger by integrating a thrombus-localized photothermal effect and on-demand nitric oxide release. This nanoplatform offers a versatile approach for the diagnosis and treatment of life-threatening diseases caused by various thrombotic disorders.

The formation of an occlusive thrombus in blood vessel is the main culprit for numerous life-threatening cardiovascular diseases, such as myocardial infarction, ischemic stroke, and pulmonary embolism, which represent the leading causes of morbidity and mortality worldwide[1–3]. At present, the treatment strategies for thrombosis mainly include the administration of thrombolytic agents such as tissue plasminogen activator, the FDA-approved thrombolytic drug for ischemic stroke, to dissolve blood clots[4,5]. However, the clinically used anti-thrombotic drugs usually offer limited therapeutic benefit due to the short circulation half-life (only a few minutes), narrow therapeutic time window (within a few hours after onset of symptoms), low targeting ability (<5% of the drug reaching lesion site), and inferior

[1]State Key Laboratory of Medicinal Chemical Biology, Key Laboratory of Bioactive Materials, Ministry of Education, Frontiers Science Center for Cell Responses, and College of Life Sciences, Nankai University, Tianjin 300071, China. [2]Tianjin Key Laboratory of Biomedical Materials and Key Laboratory of Biomaterials and Nanotechnology for Cancer Immunotherapy, Institute of Biomedical Engineering, Chinese Academy of Medical Sciences and Peking Union Medical College, Tianjin 300192, China. ✉e-mail: kongdeling@nankai.edu.cn; liwen@bme.pumc.edu.cn; qiji@nankai.edu.cn

thrombus penetration[6,7]. High doses of thrombolytic drugs are compelled to be used to achieve efficient thrombolysis, which, however, increases the risk of undesirable bleeding complications[8,9]. Therefore, rational design of novel strategies with high efficiency and security has been a high priority for thrombotic disease treatment.

Early and accurate diagnosis of thrombus is of tremendous importance for timely treatment of thrombus as well as prevention of its life-threatening complications[10,11]. Traditional imaging modalities, such as magnetic resonance imaging (MRI), and the ionizing radiation-based computed tomography (CT) and positron emission tomography (PET), have been used for detecting thrombus-related diseases, especially the extensively developed thrombus in the clinical field[12,13]. However, such imaging modalities share the drawbacks of weak functional signal, suboptimal spatial resolution, time-consuming examination processes and the requirement of bulky instruments[14,15]. In addition, for the low sensitivity and specificity, these imaging methods are usually difficult to accurately identify thrombus in the early stage and afford real-time information on the pathological state, both of which are necessary for urgently treating thrombus[16]. As an

alternative, optical imaging is a rapidly developed detection technology with the advantages of high resolution and sensitivity, real-time visualization, fast feedback and non-ionizing radiation[17–19]. With these features, optical imaging has been broadly adopted for disease detection and image-guided surgery in both preclinical and clinical research[20,21]. However, the penetration depth of traditional optical imaging is usually very limited, which severely impedes its applications for in vivo imaging and monitoring of thrombosis[22,23].

By integrating light excitation and ultrasound detection, photoacoustic (PA) imaging is an emerging optical imaging technique that has gained increasing interest recently for biomedical applications[24–26]. Compared with conventional fluorescence imaging, PA imaging has higher spatial resolution and deeper penetration depth as the scattering of acoustic waves in tissue is orders of magnitude weaker than that of light; and compared with ordinary ultrasound imaging, PA imaging has better tissue contrast, thus representing a complementary imaging modality[27,28]. More interestingly, PA imaging in the second near-infrared (NIR-II, 1000–1700 nm) window has been developed very recently. In particular, the spectral region of 1000–1350 nm is

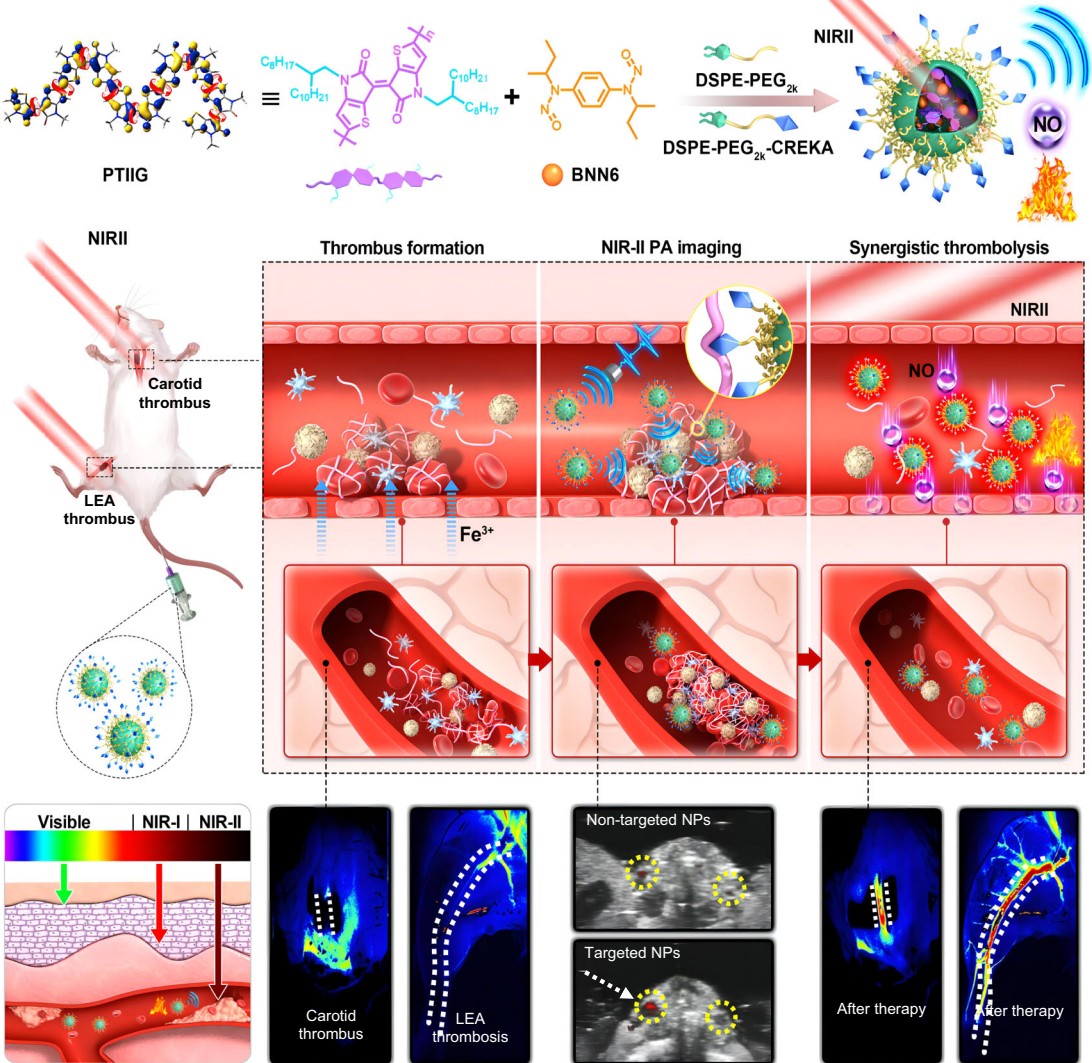

**Fig. 1 | Schematic illustration of the fibrin-specific nanoprobes for NIR-II PA imaging and photo-triggered synergistic treatment of thrombosis.** A highly NIR-II light-harvesting semiconducting homopolymer (PTIIG) and a thermo-sensitive NO donor (BNN6) were formulated into NPs, on which the fibrin-specific peptide (CREKA) was functionalized to ensure selective thrombus targeting. With the features of bright NIR-II PA signal and active thrombus accumulation ability, the nanoprobe was able to sensitively diagnose thrombus via NIR-II PA imaging. The nanoplatform also exerted potent thrombolytic properties by combining thrombus-localized photothermal effect and on-demand NO release, which remarkably accelerated thrombus removal in both carotid thrombosis model and low extremity arterial (LEA) thrombosis model.

considered to be favorable for PA imaging as the main endogenous absorbers show relatively low absorption in such spectral window and the NIR-II light also has significantly reduced light scattering in tissue and minimal background interference[29–31]. As a result, instead of using excitation wavelength in the first NIR region (NIR-I, 700–900 nm), the NIR-II light-excited PA imaging could provide higher signal-to-noise ratio (SNR) and deeper tissue penetration[32–34]. For instance, the tissue penetration depths of NIR-I and NIR-II PA imaging techniques are reported to be 2–3 cm and 5–6 cm, respectively[29,31]. These features make NIR-II PA imaging a very promising method for the diagnosis of thrombosis. Although NIR-II PA imaging has been reported for biomedical applications, most of which focused on the tumor-related diagnosis and treatment[31]. The highly effective and thrombus-targeted NIR-II PA imaging is still an area of challenge to be exploited so far. In addition, to our best knowledge, there have been no reports on the fabrication of theranostic platforms that could not only facilitate the in vivo precise diagnosis of thrombosis using NIR-II PA imaging but also exert excellent antithrombotic activity.

As antithrombotic drug usually suffers from unsatisfactory therapeutic effects and biosafety risks, nonpharmaceutical thrombolysis strategies have been proposed in recent years[35,36]. There is growing evidence showing that localized hyperthermia could accelerate the ablation of blood clot via loosening the non-covalent interactions of fibrins, providing a safe and non-invasive thrombolytic approach[37–39]. As PA signal is generated when a pulsed laser illuminates the absorber to induce local temperature rise and thermoelastic expansion to produce acoustic waves, PA property is commonly accompanied by photothermal effect in the same agent[40–42]. Nevertheless, photothermal thrombolysis alone is usually not potent enough for complete thrombus eradication, leading to high recurrence rates post-therapy. To deal with the unsatisfactory outcomes of photothermal thrombolysis, other therapeutic modalities are needed to be integrated. As an important vasoprotective molecule, nitric oxide (NO) plays a pivotal role in maintaining vascular homeostasis, inhibiting platelet activation, and regulating vasodilation[43–46]. It has been demonstrated that the endothelial cells in healthy blood vessels continue to produce low concentrations of NO to prevent thrombosis[47]. And the decreased release of endothelium-derived NO is one of the earliest and most important characteristics of endothelial dysfunction[48,49]. With the powerful vascular function regulation capability, NO represents a promising treatment option for thrombolysis and prevention of thrombosis recurrence[50–52]. However, due to the random trajectories and rapid quenching of NO by molecular oxygen, one of the main difficulties in NO therapy is the on-demand control delivery of NO to the thrombotic lesion.

In this work, we develop a theranostic nanoplatform that integrates the long-wavelength NIR-II PA imaging-based sensitive thrombosis diagnosis and excellent antithrombotic activity (Fig. 1). A kind of semiconducting homopolymer possessing distorted backbone structure is designed and synthesized, which exhibits excellent NIR-II light harvesting ability and violent molecular motion that helps to boost the photothermal conversion and PA signal. The polymer is formulated into nanoparticles (NPs), on which a fibrin-specific peptide is functionalized to ensure selective thrombus targeting. The strong NIR-II absorption and superb photo-to-heat conversion ability could amplify the PA diagnostic signal for highly sensitive thrombus detection and monitoring, and induce localized hyperthermia for non-invasive lysis of fibrin clots. Moreover, a thermosensitive NO donor, N,N'-di-sec-butyl-N,N'-dinitroso-1,4-phenylenediamine (BNN6), is loaded into the polymer matrix, from which the in situ generation of NO could be triggered by the NIR-II light irradiation-induced hyperthermia. The combination of thrombus-localized photothermal effect and controlled NO release not only exert potent thrombolytic properties but also facilitate the deep penetration of NPs into thrombus to boost antithrombotic outcomes. Meanwhile, NO is able to prevent thrombus

recurrence by inhibiting platelet activation and aggregation. The nanoprobes not only display high–contrast NIR-II PA signal in the obstructed vessels, but also remarkably accelerate thrombus eradication in both carotid thrombosis model and low extremity arterial thrombosis model, eliciting a substantially improved therapeutic outcomes over free antithrombotic drugs. The nanoplatform possesses good in vivo biosafety with a low risk of bleeding complications and no detectable damage to the blood vessels. These features make the NIR-II theranostic nanoplatform developed in this work promising for potential clinical translation, and we hope it will give inspirations for the further development of diagnosis and treatment strategy for life-threatening diseases caused by various thrombotic disorders.

## Results

### Design, synthesis, and characterization of PTIIG

Among various PA agents, semiconducting polymers (SPs) have received increasing attention for the advantages of high photostability, good biocompatibility, and easy chemical modification to provide desired photophysical properties[53–55]. However, the absorption of organic/polymer agents in NIR region, especially in the long-wavelength NIR-II window, is usually insufficient to generate strong NIR-II PA signal. Herein, we develop a high-performance NIR-II polymer PA probe via the rational molecular design. The donor-acceptor (D-A) approach that the electron-donating and -withdrawing units are alternatively conjugated to yield a copolymer represents a widely used strategy to tune the bandgap of SPs and realize NIR-II absorption. Nevertheless, the electrostatic attraction of D and A moieties in D-A copolymer usually leads to strong intermolecular interaction, thus severely restricts the molecular motion in aggregate state[56,57]. As molecular motion plays an important role in amplifying the PA signal, the homopolymers consisting of one building block would be an alternative option as they could efficiently avoid the intermolecular interaction due to the electron repelling effect (Fig. 2a)[58,59]. In this work, we designed and synthesized an ultrasmall-bandgap semiconducting homopolymer with strong NIR-II absorption and violent molecular motion. As depicted in Fig. 2b and Supplementary Fig. 1, the dibromo thienoisoindigo (TIIG) monomer was polymerized with hexamethylditin via Stille cross–coupling reaction, which resulted in PTIIG homopolymer. TIIG is a highly conjugated structure with enhanced electron delocalization along the backbone. The long branched aliphatic chains would endow the resulting polymers with good solubility, and provide some space for molecular motion in the aggregate state. The polymer was characterized by gel permeation chromatography (GPC), which gave a number-average molecular weight ($M_n$) and polydispersity index (PDI) of 32.1 kDa and 1.73, respectively. PTIIG also exhibited good processability as it could be facilely dissolved in common solvents such as tetrahydrofuran (THF), chloroform and toluene.

In order to gain in-depth understanding about the molecular structure and electronic property, density functional theory (DFT) calculation was conducted to study the oligomers with different numbers of repeat units. As shown in Fig. 2c, the optimized molecular geometries of TIIG1–5 suggested rather twisted structures, and the backbones possessed crowded 3D conformation. Accordingly, intensive molecular motion would occur, which was expected to be beneficial for boosting PA signal. Interestingly, the electron cloud in all the oligomers was distributed along the whole molecular backbone, indicating excellent conjugation. Moreover, there was nearly no charge separation between the highest occupied molecular orbital (HOMO) and lowest unoccupied molecular orbital (LUMO) in this homopolymer, which was quite different from D-A copolymer that strong intramolecular charge transfer usually existed[60]. The energy levels of TIIG1–5 were also calculated. Noteworthy, the electronic bandgap significantly decreased as the molecular length increased (Supplementary Fig. 2), suggesting that the homopolymer

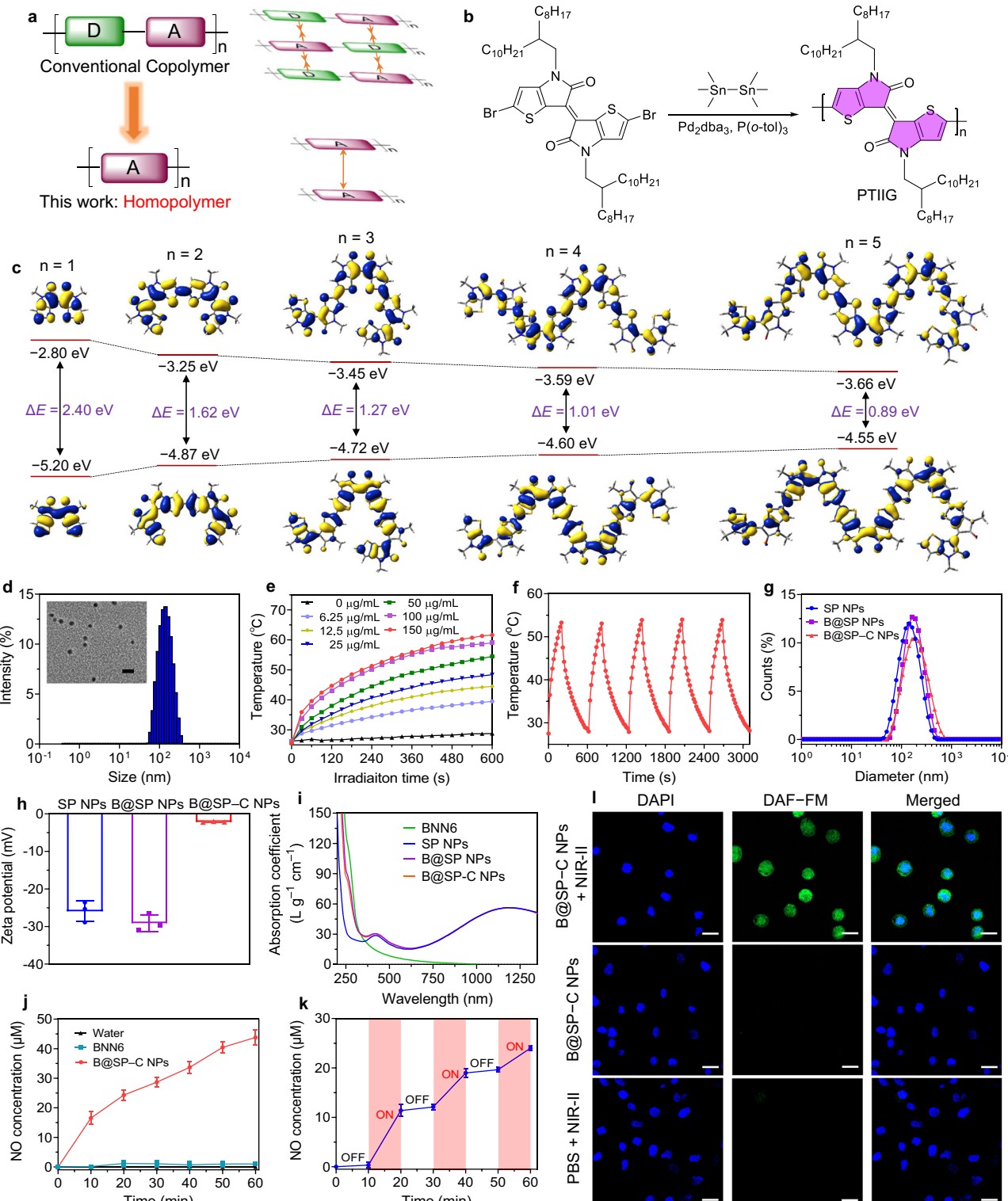

**Fig. 2 | Characterization of PTIIG polymer and NPs. a** Schematic of the backbone structures and intermolecular interaction of conventional D-A copolymer and the homopolymer in this work. **b** Synthetic route to PTIIG. **c** Theoretical calculation of the molecular geometries, HOMO, LUMO electron cloud distribution and electronic bandgaps of PTIIG with different numbers of repeat units (1–5).
**d** Representative DLS analysis and TEM image of SP NPs. Scale bar = 250 nm. **e** The photothermal heating curves of SP NPs at different concentrations (0–150 µg mL$^{-1}$ based on PTIIG) under 1064 nm light (1 W cm$^{-2}$) irradiation. **f** The photothermal stability of SP NPs during five circles of laser on-off processes. **g** The size distribution and **h** zeta potential of SP NPs, B@SP NPs, and B@SP-C NPs as analyzed by DLS. Data are presented as mean ± SD ($n$ = 3 independent experiments). **i** The

absorption spectra of BNN6, SP NPs, B@SP NPs, and B@SP-C NPs at the same concentration based on PTIIG (6 µg mL$^{-1}$). **j** The amount of NO release from water, BNN6, and B@SP-C NPs under 1064 nm (1 W cm$^{-2}$) light irradiation. Data are presented as mean ± SD ($n$ = 3 independent experiments). **k** The pulsatile NO release triggered by alternately adjusting the NIR-II laser between "on" and "off" states. Data are presented as mean ± SD ($n$ = 3 independent experiments). **l** Representative CLSM images of HUVECs treated with PBS + NIR-II laser, B@SP-C NPs, or B@SP-C NPs + NIR-II laser, respectively. Nuclei were stained with DAPI (blue signal), and NO was stained with DAF-FM DA probe (green signal). Scale bars = 50 µm. For **d**–**g**, **i**, **l**, experiment was repeated three times independently with similar results. Source data are provided as a Source Data file.

might possess a rather small bandgap. The absorption spectrum of PTIIG in THF was presented in Supplementary Fig. 3, which showed maximal absorption at 1310 nm. Interestingly, different from most D-A copolymers that possessed a strong absorption peak in UV spectral region and a relatively weak band in long-wavelength range, the homopolymer in this work exhibited broad and intense NIR-II absorption, which was around 3-fold higher than the absorption peak at about 400 nm. Based on the onset of absorption spectrum, the electronic bandgap of PTIIG was calculated to be as small as 0.65 eV, which was among the smallest bandgaps of SPs[61]. Thanks to the extended conjugation length mediated by electron delocalization along the molecular backbone, the homopolymer exhibited very strong absorption in NIR-II region with molar absorption coefficient as high as $57.2 \, L \, g^{-1} \, cm^{-1}$ at 1300 nm. These results indicated that PTIIG was a highly light-absorbing NIR-II polymer, holding great potential for PA and photothermal applications. X-ray diffraction analysis of PTIIG film indicated that the homopolymer didn't have noticeable crystalline signature (Supplementary Fig. 4), in which the loose molecular packing was expected to facilitate the molecular motion and photo-to-acoustic conversion[58].

## Fabrication and characterization of NPs

The polymer PTIIG was readily assembled into water-soluble nanoparticles (NPs) via the one-step nanoprecipitation method with 1,2-distearoyl-*sn*-glycero-3-phosphoethanolamine-*N*-methoxy(poly ethylene glycol)−2000 (DSPE-PEG$_{2000}$) as the surfactant. Nanoprecipitation is a widely used method to formulate NPs, during which the hydrophobic organic molecules randomly assemble in the core, and the amphiphilic surfactant forms the shell layer coated on NP surface[62]. Dynamic light scattering (DLS) and transmission electron microscope (TEM) measurements were conducted to characterize the size and morphology. As shown in Fig. 2d, DLS results revealed that the NPs obtained from PTIIG (designated SP NPs) possessed an average diameter of 126 nm and a PDI of 0.15, and TEM image suggested that SP NPs had uniform spherical structure with an average diameter of about 110 nm. Compared with the hydrodynamic diameter determined by DLS, the relatively small diameter from TEM measurement was likely due to the drying and shrinkage of NPs during the sample preparation[63]. SP NPs exhibited similar absorption profile as that in solution state (Supplementary Fig. 5), with strong absorption in the spectral region of 1000−1400 nm. The strong absorption of SP NPs in NIR-II region motivated us to investigate their NIR-II photothermal effect. Upon 1064 nm laser irradiation, SP NPs showed rapid temperature increase and reached a plateau in 10 min, while the temperature of PBS solution barely changed in the same condition. The amplitude of photo-induced temperature elevation increased with the concentrations of NPs and the laser power intensity (Fig. 2e, Supplementary Fig. 6), and the temperature could rise to near 60 °C when 0.1 mg mL$^{-1}$ (based on PTIIG) of SP NPs were irradiated with 1064 nm light (1 W cm$^{-2}$) for 10 min. SP NPs were found to possess excellent photostability, as their photothermal property remained unchanged during five circles of laser on-off processes (Fig. 2f). According to Jablonski diagram, there are several competitive pathways including radiative decay, nonradiative deactivation, and electron transfer to the triplet excited state for generating reactive oxygen species (ROS) that may consume the excited-state energy[64]. We therefore studied the related photophysical processes of SP NPs to explore their main energy dissipation pathway. The photoluminescence spectrum of SP NPs was first analyzed, which showed no detectable signal (Supplementary Fig. 7). In addition, there was also no observed ROS production under light irradiation. These results suggested that the excited state energy of SP NPs mainly concentrated on thermal deactivation process, being beneficial for photothermal and PA applications.

After confirming the excellent NIR-II light harvesting ability and photothermal effect of SP NPs, we then sought to construct a NO donor-loaded nanoplatform for simultaneous thrombus imaging and treatment. BNN6, a temperature-sensitive NO donor, was prepared (the synthesis route and characterization were presented in Supplementary Figs. 8, 9) and doped into PTIIG matrix to form B@SP NPs, from which the on-demand NO release was expected to be triggered by NIR-II light-induced local temperature rise[51]. To endow the NPs with thrombus-homing ability, a fibrin-specific peptide, Cys-Arg-Glu-Lys-Ala (CREKA), was conjugated with DSPE-PEG$_{2000}$-Mal to obtain DSPE-PEG$_{2000}$-CREKA (Supplementary Fig. 10), which was further decorated onto the NPs via the insertion of hydrophobic DSPE section into the polymer matrix. During thrombus progression, fibrin is generated upon the activation of coagulation cascades, then deposited both inside and on the surface of thrombus to stabilize it, thus fibrin is an important target for site-specific delivery of thrombolytic agents to thrombi[65,66]. The resulting B@SP−C NPs showed uniform spherical morphology, with an average hydrodynamic diameter (168 nm) slightly larger than that of non-targeted B@SP NPs (151 nm) (Fig. 2g, Supplementary Fig. 11). Upon the modification with CREKA peptide, the zeta potential of NPs changed from −29 mV to −2 mV, probably due to the positively charged feature of CREKA (Fig. 2h). All the nanoformulations including SP NPs, B@SP NPs and B@SP−C NPs exhibited similar appearance in water as clear olive solution (Supplementary Fig. 12). The UV-vis-NIR absorption spectrum of B@SP−C NPs displayed a broad absorption band with maximum at around 1200 nm, which was consistent with that of PTIIG polymer. In addition, a new absorption peak at 258 nm was observed, typically attributed to the signal of BNN6 (Fig. 2i). Noteworthy, the photothermal conversion efficiency (PCE) of B@SP−C NPs was calculated to be as high as 73% (Supplementary Fig. 13), which was among the highest values of SPs, especially for NIR-II agents. The high PCE could be attributed to the intensive molecular motion of the twisted polymer backbone of PTIIG. According to the absorption spectra of BNN6 standard solutions, the encapsulation efficiency of BNN6 in B@SP−C NPs was determined as 41.2% (Supplementary Fig. 14). Fourier transform infrared spectroscopy (FT-IR) was further used to characterize the chemical composition of B@SP−C NPs. As illustrated in Supplementary Fig. 15, the characteristic bands of BNN6 and CREKA peptide appeared in the FT-IR spectrum of B@SP−C NPs, indicating the successful introduction of NO donor and targeting peptide. In addition, after incubating the 1,1-dioctadecyl-3,3,3′,3′-tetramethylindocarbocyanine perchlorate (DiI)-loaded B@SP−C NPs with FITC-labeled fibrin, good colocalization of the red fluorescence from NPs with the green fluorescence from fibrin was observed (Supplementary Fig. 16), indicating the specific binding between them. Moreover, B@SP−C NPs exhibited good colloidal stability in dark condition at 4 °C for one week, with no apparent changes in the particle size (Supplementary Fig. 17). We also measured the size, PDI, and compound encapsulation efficiency of B@SP−C NPs prepared from different batches, and the results suggested that the NPs had good reproducibility (Supplementary Fig. 18).

## Photo-triggered NO release and PA property

Following the fabrication of B@SP−C NPs, we then studied the NIR-II light-triggered NO release behavior. First, the loading of BNN6 and modification of CREKA peptide was confirmed to have no apparent impact on the photothermal effect of NPs (Supplementary Fig. 19). To verify whether the localized hyperthermia of NPs could activate temperature-sensitive BNN6 to generate NO, the NO release under 1064 nm laser irradiation (1 W cm$^{-2}$) was quantitatively detected using Griess assay. As presented in Fig. 2j, BNN6 solution alone exhibited only very slight NO release under NIR-II light exposure. In sharp contrast, B@SP−C NPs were able to continuously produce high level of NO in response to NIR-II light exposure. Moreover, the NO generation from B@SP−C NPs exhibited a dose-dependent manner, as more NO was detected when the concentration of B@SP−C NPs increased (Supplementary Fig. 20). The on-demand and pulsatile NO release

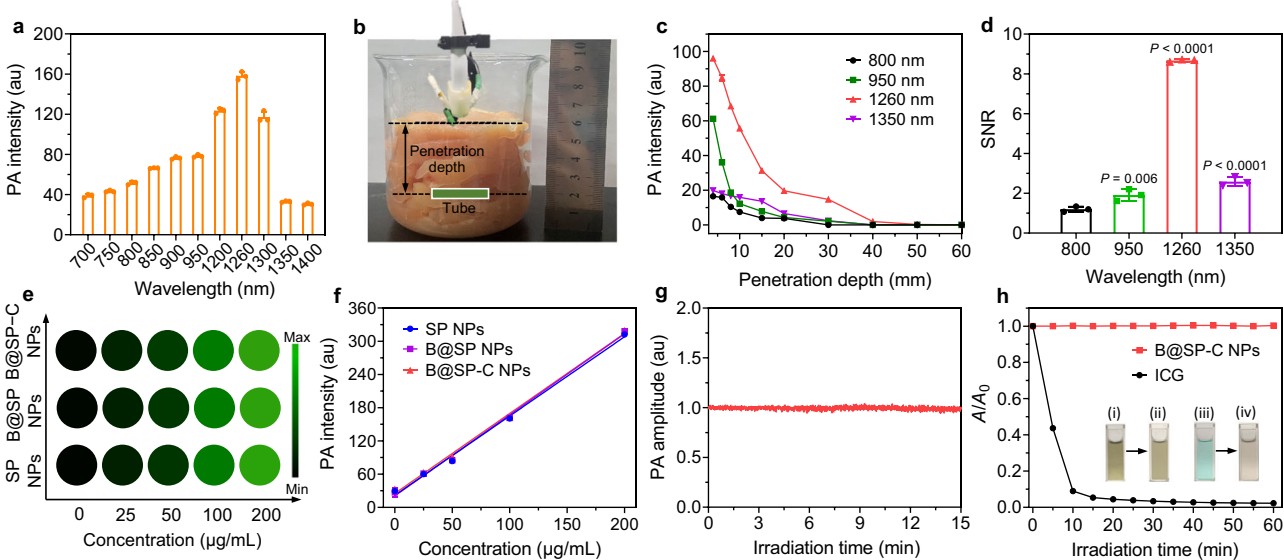

**Fig. 3 | Investigation of in vitro PA property. a** The PA intensity of B@SP−C NPs (100 μg mL⁻¹ based on PTIIG) with the excitation of various wavelengths of light. Data are presented as mean ± SD ($n$ = 3 independent experiments). **b** A photograph showing the measurement of PA signal of B@SP−C NPs underneath check breast. **c** PA intensities of B@SP−C NPs (100 μg mL⁻¹ based on PTIIG) under various thicknesses of chicken breast upon 800, 950, 1260, or 1350 nm laser irradiation. Data are presented as mean ± SD ($n$ = 3 independent experiments). **d** The SNR of PA images of B@SP−C NPs under 10 mm of chicken breast upon 800, 950, 1260, or 1350 nm laser irradiation. Data are presented as mean ± SD ($n$ = 3 independent experiments). $P$ values were calculated using one-way ANOVA for multiple comparisons. **e** PA images and **f** the corresponding PA amplitudes of different

concentrations of SP NPs, B@SP NPs, and B@SP−C NPs under 1260 nm irradiation. Data are presented as mean ± SD ($n$ = 3 independent experiments). **g** PA intensity of B@SP−C NPs upon 1260 nm laser exposure for 15 min. **h** Plots of $A/A_0$ of B@SP−C NPs and ICG under laser irradiation for different time. $A_0$ and $A$ are the absorption intensity of B@SP−C NPs (at 1260 nm) or ICG (at 780 nm, the maximal absorption of ICG) before and after laser irradiation, respectively. B@SP−C NPs and ICG solution were irradiated with continuous 1064 nm (1.0 W cm⁻²) and 808 nm (1.0 W cm⁻²) lasers, respectively. Inset shows the photographs of (i, ii) B@SP−C NPs and (iii, iv) ICG solution before and after light irradiation for 60 min. Source data are provided as a Source Data file.

could also be achieved by alternately adjusting the NIR laser between "on" and "off" states (Fig. 2k). These results demonstrated that the NIR-II light-activated nanosystem allowed for the spatiotemporal controlled NO delivery, which might greatly enhance the safety and efficiency of NO gas-mediated thrombolysis. The light-triggered NO release was further assessed in cell culture condition (Fig. 2l). The human umbilical vein endothelial cells (HUVECs) were treated with PBS + NIR-II light, B@SP−C NPs, or B@SP−C NPs + NIR-II light, respectively. 4-Amino-5-methylamino-2′,7′-difluorofluorescein diacetate (DAF-FM DA) was used as the NO indicator, which could cross the plasma membrane and react with NO to yield the green fluorescent triazolofluorescein[67]. According to the confocal laser scanning microscope (CLSM) images, the cells treated with B@SP−C NPs plus NIR-II light exhibited bright green fluorescence signal, whereas no noticeable fluorescence was observed from the cells treated with B@SP−C NPs or light irradiation alone.

Subsequently, the potential of B@SP−C NPs for thrombus theranostics was explored. For this purpose, the in vitro NIR-II PA signal of NPs was first measured. The optimum excitation wavelength was determined by measuring the PA amplitude in different wavelengths. The PA signal of B@SP−C NPs (Fig. 3a) matched well with their absorption property, indicating that the PA signal indeed originated from the NIR absorption of PTIIG. To evaluate the penetration depth of different wavelengths light for PA imaging, B@SP−C NPs embedded underneath various thicknesses of chicken breast were irradiated with 800, 950, 1260, or 1350 nm pulsed lasers, and the PA amplitude was recorded (Fig. 3b). As revealed in Fig. 3c, the PA signals of B@SP−C NPs under 1260 nm laser irradiation were obviously stronger than that excited by other wavelengths at all the tested tissue thicknesses. Interestingly, the PA images at 1260 nm laser irradiation also displayed a pronounced SNR value of 8.7 in the thickness of 10 mm, which was 7.2-, 4.6- and 3.4-fold higher than that of 800, 950 and 1350 nm laser

irradiation, respectively (Fig. 3d). The excellent PA imaging performance of B@SP−C NPs under 1260 nm laser was likely ascribed to the reduced light scattering and attenuation of 1260 nm light through tissue than 800 nm and 950 nm light, and the 1260 nm light was also reported to be less absorbed by water when compared to 1350 nm light. Next the PA intensities of B@SP−C NPs at different concentrations were recorded, which revealed a good linear correlation between the PA amplitude and NPs concentration, suggesting its potential for quantitative analysis (Fig. 3e, f). The stability of NIR-II PA signal was then evaluated by scanning the NPs with 1260 nm pulsed laser for 15 min, and no signal attenuation was observed after laser exposure (Fig. 3g). Under the irradiation of continuous 1064 nm NIR-II light for 60 min, B@SP−C NPs showed good stability and nearly no decrease in the absorption intensity. In contrast, the clinically used indocyanine green (ICG) displayed significant signal decrease after laser irradiation and its NIR absorption nearly disappeared in 20 min (Fig. 3h, Supplementary Fig. 21). These results demonstrated that B@SP−C NPs not only allowed for highly sensitive PA imaging but also possessed good stability and was suitable for long-term PA diagnosis.

**Thrombus targeting and thrombolytic capability in vitro**

Encouraged by the excellent photothermal property and controlled NO release behavior of B@SP−C NPs, we next explored the thrombolytic capability. The in vitro biosafety of B@SP−C NPs was first evaluated by co-culturing NPs with NIH 3T3 cells or HUVEC cells for 24 h. Then the cell viability was analyzed using 3-(4,5-dimethylthiazil-2-yl)−2,5-diphenyltetrazolium bromide (MTT) assay, which showed that more than 90% of cells maintained alive after treating with 0−200 μg mL⁻¹ of nanoprobe (Supplementary Fig. 22), revealing the negligible cytotoxicity. The blood compatibility of B@SP−C NPs was also tested by hemolysis experiment (Fig. 4a). With the concentrations of B@SP−C NPs ranging from 0 to 200 μg mL⁻¹, all the blood samples

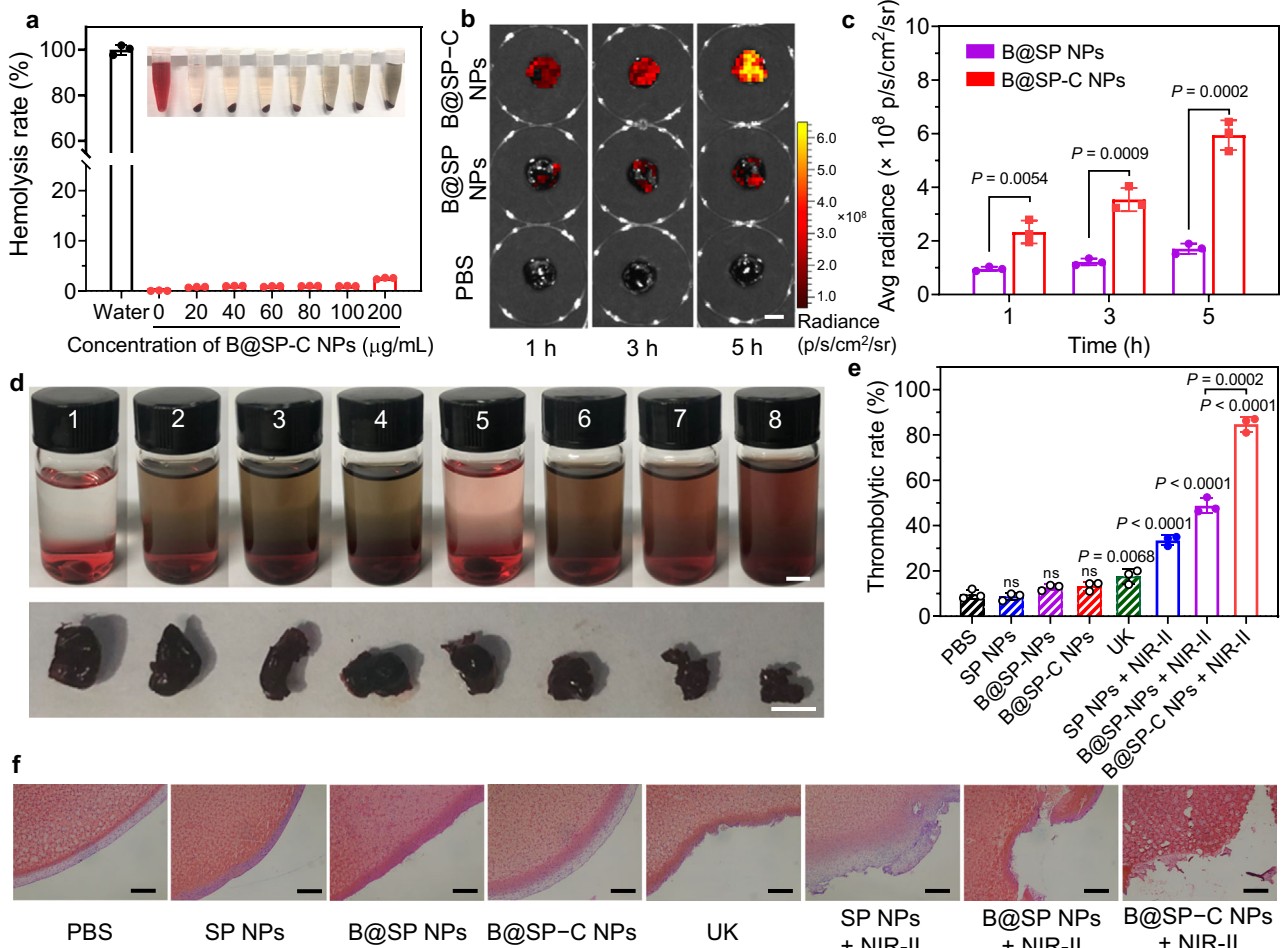

**Fig. 4 | Thrombus targeting and thrombolytic capability in vitro. a** The hemolysis rate of red blood cells treated with water and different concentrations of B@SP–C NPs. Inset shows the red blood cells solution upon different treatments (from left to right: water, 0, 20, 40, 60, 80, 100, and 200 μg mL⁻¹ of B@SP–C NPs). Data are presented as mean ± SD ($n$ = 3 independent experiments). **b** The fluorescence images of artificial blood clots after incubation with PBS, B@SP NPs, or B@SP–C NPs. The NPs were labeled with DiR (red signal). Scale bar = 5 mm. **c** Quantitative analysis of the relative fluorescence intensity of the artificial blood clots in (**b**). Data are presented as mean ± SD ($n$ = 3 independent experiments). $P$ values were calculated using two-tailed Student's $t$-test. **d** Photographs of the blood clot solution after different treatments (from left to right: PBS, SP NPs, B@SP NPs, B@SP–C NPs, free UK, SP NPs + NIR-II laser, B@SP NPs + NIR-II laser, and B@SP–C NPs + NIR-II laser). The residual blood clots were taken out and shown below. Scale bars = 5 mm. **e** Thrombolytic rates were quantitatively analyzed by measuring the percentage of thrombus mass loss after different treatments in **d**. Data are presented as mean ± SD ($n$ = 3 independent experiments). $P$ values were calculated using one-way ANOVA for multiple comparisons. **f** H&E staining of the residual blood clots after different treatments. Scale bars = 100 μm. Experiment was repeated three times independently with similar results. Source data are provided as a Source Data file.

showed relatively low hemolysis rates of <3%, indicating the good hemocompatibility. To start the thrombolytic test, the targeting specificity of B@SP–C NPs to thrombus was examined in ex vivo blood clots. The artificial blood clots were prepared by mixing fresh mice blood with thrombin, and the NPs were labeled with a NIR dye, 1,1′-dioctadecyl-3,3,3′,3′-tetramethylindotricarbocyanine iodide (DiR) to be visualized by fluorescence imaging. After the incubation of artificial clots with DiR-loaded B@SP–C NPs or non-targeted B@SP NPs for 1 h, 3 h or 5 h, the clots were imaged under in vivo imaging system (IVIS). As expected, the blood clots incubated with B@SP–C NPs exhibited remarkably stronger red fluorescence than B@SP NPs and PBS groups at each time point (Fig. 4b, c), which manifested improved targeting property mediated by CREKA peptide. Then the thrombolytic effects of various formulations were investigated by subjecting the blood clots to different treatments, i.e., (1) PBS, (2) SP NPs, (3) B@SP NPs, (4) B@SP–C NPs, (5) urokinase (UK), (6) SP NPs + NIR-II light, (7) B@SP NPs + NIR-II light, and (8) B@SP–C NPs + NIR-II light. In all the laser-treated groups, the blood clots were illuminated with 1064 nm NIR-II light (1 W cm⁻²) for 20 min. After exposure to B@SP–C NPs + NIR-II light, the

size of blood clots significantly shrank and the supernatant turned to blood-red color (Fig. 4d). In contrast, the blood blots in other groups only showed slight or moderate dissolution. The thrombolytic efficiency was quantitatively analyzed by measuring the percentage of thrombus mass loss after different treatments. The thrombolytic rates of SP NPs + NIR-II, B@SP NPs + NIR-II, and B@SP–C NPs + NIR-II sequentially increased from 33% to 49%, and to 85% (Fig. 4e), affirming that both the NIR light-induced photothermal/NO production and homing peptide modification could amplify the thrombolysis outcomes. Meanwhile, the photothermal effect on thrombus lysis was verified by comparing the clot dissolution efficiency of SP NPs (8.5%) and SP NPs + NIR-II light (33%). It was noted that the B@SP–C NPs + NIR-II light group also demonstrated far stronger thrombolytic ability than UK, the clinical thrombolytic drug. Furthermore, the residual blood clots were sectioned for hematoxylin and eosin (H&E) staining to show the clot-lysing process (Fig. 4f). For the PBS and NPs alone-incubated groups, the blood clots were almost intact with smooth boundaries. Nevertheless, the surface of the blood clots treated with UK, SP NPs + NIR-II or B@SP NPs + NIR-II became rough and

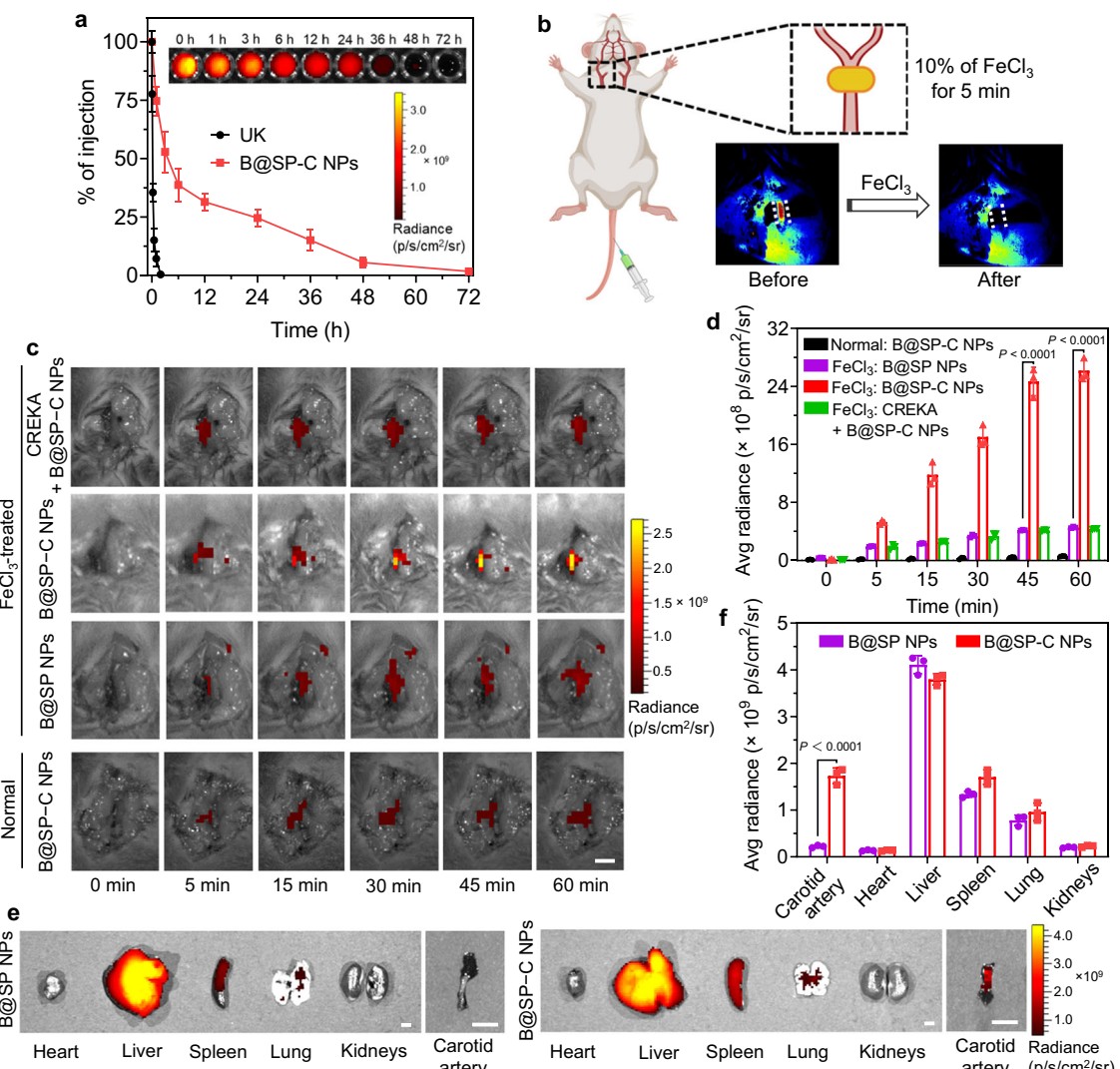

**Fig. 5 | In vivo thrombosis-targeting ability. a** Blood circulation profile of UK and B@SP-C NPs after intravenous injection into mice. Inset shows the fluorescence images of blood drawn from the mice at the indicated time points after B@SP-C NPs injection. Data are presented as mean ± SD (*n* = 3 mice). **b** Scheme of establishing the FeCl₃-induced carotid arterial thrombosis model. Inset shows the LSBFMS images before and after thrombus formation. The illustration was created with the help of BioRender.com. **c** Representative in vivo fluorescence images of the healthy normal carotid artery and the thrombotic artery at different time points post-injection of B@SP NPs, B@SP-C NPs, or CREKA + B@SP-C NPs. Scale bar = 3 mm. **d** Quantitative analysis of the fluorescence intensity of the carotid arteries in (**c**). Data are presented as mean ± SD (*n* = 3 mice). *P* values were calculated using one-way ANOVA for multiple comparisons. **e** Representative ex vivo fluorescence images of the major organs and carotid thrombotic artery of mice collected at 2 h after B@SP NPs or B@SP-C NPs injection. Scale bars = 3 mm. **f** Quantification analysis of the fluorescence intensity of different organs and carotid artery. Data are presented as mean ± SD (*n* = 3 mice). Statistical significance was calculated using two-tailed Student's *t*-test. Source data are provided as a Source Data file.

damaged, and the B@SP−C NPs + NIR-II light group again showed the most potent thrombolysis effect with the blood clots degraded stepwise into many small pieces.

## In vivo targeting and NIR-II PA imaging of thrombus

We then tested the feasibility of B@SP-C NPs for thrombus theranostics in living mice. The systemic circulation time of therapeutic agents has significant impact on the treatment outcomes, especially for the lesions located in blood vessels. Particularly, we expected that the PEGylation and CREKA peptide decoration would not only prolong the circulation time of NPs in blood but also facilitate the thrombus-specific accumulation of NPs. To confirm our hypothesis, in vivo circulation profiles of different NPs formulations were studied. The DiR-loaded NPs were injected into mice intravenously, and then blood was drawn at the designated time points to measure the fluorescence signal of NPs in blood. Unlike free UK that was quickly cleared from the

blood with a circulation half-life of ~15 min, B@SP−C NPs showed remarkably extended circulation time with a half-time of about 12 h (Fig. 5a). The prolonged circulation profile of B@SP−C NPs could be ascribed to the good colloidal stability and stealth effect of PEGylation, which would enable NPs to have greater tendency to accumulate and maintain the therapeutic effect at thrombus site. Most of B@SP−C NPs were cleared from blood at 48 h after intravenous administration, and we also found that B@SP−C NPs could be nearly completely excreted from the mouse body through biliary pathway after 7 days (Supplementary Fig. 23).

Thereafter, we evaluated the thrombus-targeting capability of NPs in vivo. For this purpose, the ferric chloride (FeCl₃)-induced carotid thrombosis model was constructed, which was based on the oxidative damage of vascular endothelial cells triggered by FeCl₃. This model has been widely used to study thrombogenesis and screen anti-thrombotic drugs[68]. As shown in Fig. 5b, a small piece of

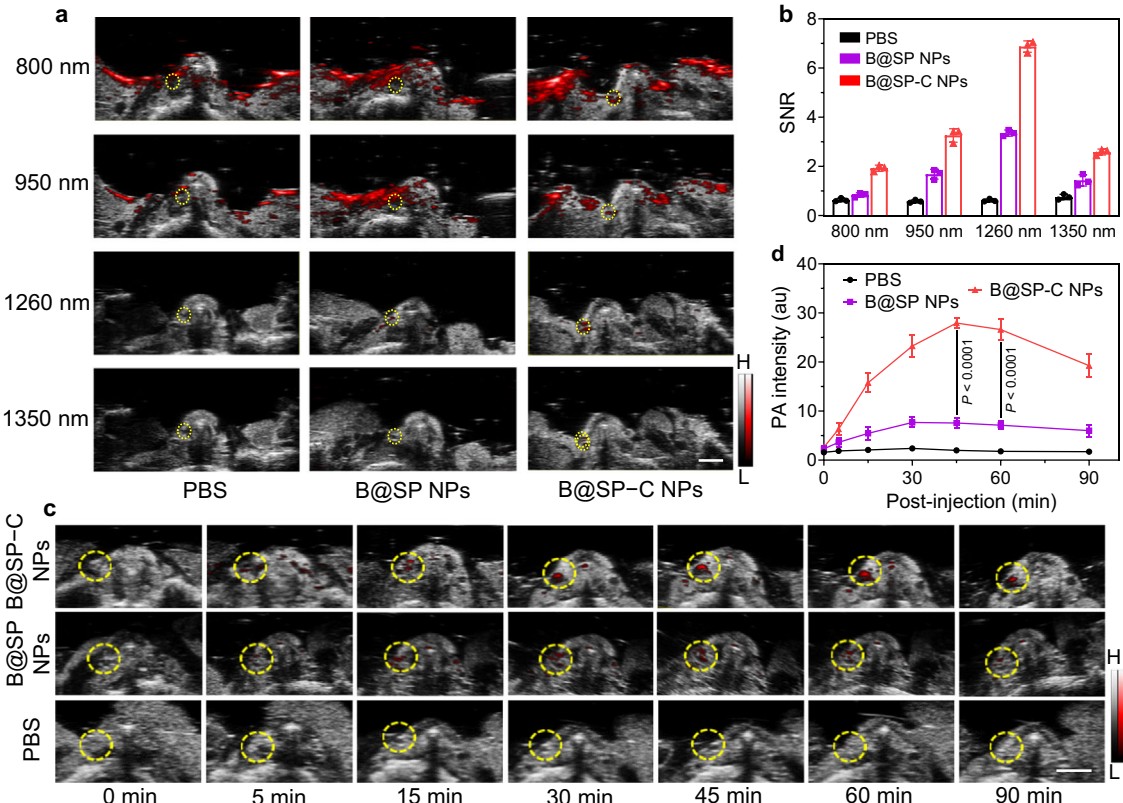

**Fig. 6 | In vivo NIR-II PA imaging of thrombus. a** Representative in vivo PA/US merged images of thrombus site with the excitation of 800, 950, 1260 and 1350 nm laser after the mice were injected with different formulations. These images were obtained by superimposing each PA image in pseudo-color on the gray-scale US image. The grey and red bars represent ultrasound and PA signals, respectively. Scale bar = 2 mm. **b** The corresponding SNR of PA signals quantified according to the images in **a**. Data are presented as mean ± SD ($n$ = 3 mice). **c** Representative in vivo PA/US merged images of the thrombotic artery at different time points as indicated after the injection of PBS, B@SP NPs or B@SP-C NPs to mice. The PA imaging was conducted with the excitation of 1260 nm light. The grey and red bars represent ultrasound and PA signals, respectively. Yellow circle indicated the thrombotic artery. Scale bar = 2 mm. **d** The corresponding PA intensity of the thrombotic artery quantified according to the images in **c**. Data are presented as mean ± SD ($n$ = 5 mice). Statistical significance was calculated using one-way ANOVA for multiple comparisons. Source data are provided as a Source Data file.

FeCl$_3$-soaked filter paper was topically applied on the carotid artery of mice for 5 min. The laser speckle blood flow monitoring system (LSBFMS) images clearly demonstrated the formation of thrombus with the closedown of blood perfusion. After the thrombosis induction, B@SP NPs or B@SP−C NPs were intravenously administrated into mice via tail vein. Then the fluorescence signal of carotid artery was acquired at different time points using IVIS. Obvious fluorescence signal was observed from the FeCl$_3$-treated artery as early as 5 min after the administration of B@SP−C NPs, and the fluorescence intensity gradually strengthened and reached maximum at 45−60 min due to the enrichment of the circulating B@SP−C NPs to thrombus site (Fig. 5c, d). Impressively, the signal intensity of B@SP−C NPs in the occlusive artery was about 6-fold stronger than that of non-targeted B@SP NPs. To figure out the mechanism behind the superior thrombus targeting ability of B@SP−C NPs, free CREKA was administrated 5 min prior to B@SP−C NPs injection to occupy the fibrin binding sites, for which the fluorescence signal in thrombus was found to be dramatically diminished, suggesting that the binding of B@SP−C NPs with thrombus was mediated via the fibrin-targeting CREKA ligand. In addition, for the non-injured control carotid artery, the injection of B@SP−C NPs only generated very weak background fluorescence, and the signal faded over time as B@SP−C NPs had little affinity to the non-injured vessel and were thus fast cleared from blood. The concentration of B@SP−C NPs in thrombosis site was calculated to be about 400 ng mg$^{-1}$, which was much higher than that in normal blood vessels (<20 ng mg$^{-1}$) (Supplementary Fig. 24). The distribution of different NPs in the major organs was also assessed.

The mice were sacrificed at 2 h after NPs administration, and carotid artery and major organs of mice including liver, lung, heart, spleen, and kidneys, were harvested. The IVIS images indicated that B@SP NPs and B@SP−C NPs exhibited similar tissue distribution patterns with a high uptake in the reticuloendothelial system (including liver and spleen) and little uptake in heart and kidneys (Fig. 5e, f). Of note, the proportion of B@SP−C NPs in the injured artery was significantly greater than that of the non-targeted B@SP NPs.

Encouraged by the excellent thrombus-targeting ability of B@SP−C NPs, we next performed in vivo NIR-II PA imaging of thrombus. Herein, the right carotid artery of mouse was treated with FeCl$_3$ to induce thrombus (the yellow circled area), whereas the contralateral carotid artery was left untreated to serve as control. The in vivo PA imaging of carotid thrombosis-bearing mice under the excitation of different wavelengths was first performed. As indicated in the representative images in Fig. 6a, there was no noticeable NIR-II PA signal in the FeCl$_3$-treated artery injected with PBS, indicating the ineffectiveness to identify thrombus without using PA contrast agent. The B@SP−C NPs administration resulted in the strongest PA signal in thrombotic artery under all the tested excitation wavelengths (Fig. 6a, b). Moreover, the PA images obtained using 1260 nm wavelength laser displayed the highest SNR in comparison with 800, 950 and 1350 nm, which was consistent with the in vitro PA results. Spectral unmixing was then performed on the PA images at different wavelengths to figure out the origin of PA signals. As shown in Supplementary Fig. 25, the strong surface background signals in the PA images under 800 and 950 nm light excitation were mainly from deoxyhemoglobin (Hb) and

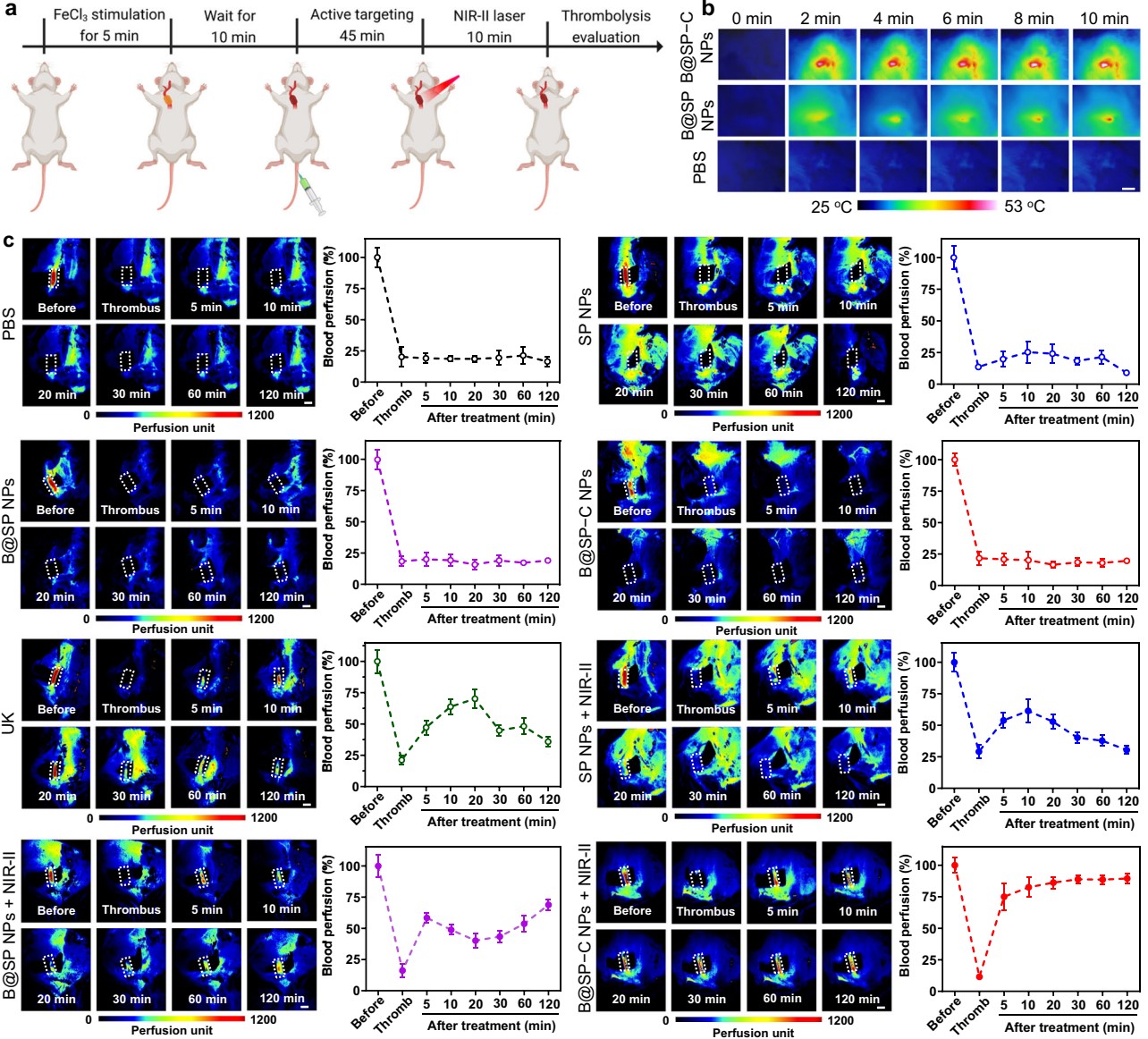

**Fig. 7 | In vivo thrombolysis evaluation in carotid arterial thrombus model.**
**a** Experimental scheme of the thrombosis model construction, antithrombotic treatment, and thrombolytic effect evaluation. The illustration was created with BioRender.com. **b** Thermal images of the thrombus site after 1064 nm NIR-II laser (1.0 W cm$^{-2}$) irradiation for different time. Scale bar = 5 mm. **c** Representative LSBFMS images and the corresponding relative blood perfusion of the mouse carotid artery after FeCl$_3$ induction and different therapeutic treatments, including PBS, SP NPs, B@SP NPs, B@SP-C NPs, free UK, SP NPs + NIR-II laser, B@SP NPs + NIR-II laser, and B@SP-C NPs + NIR-II laser. During LSBFMS imaging, the right

carotid thrombotic artery of each mouse (indicated with dotted white lines) was focused and its blood perfusion was quantified. The signals outside the carotid arteries in LSBFMS images stemmed from the blood perfusion of the surrounding capillaries. For the non-focused surrounding tissues, they might exhibit different blood perfusion signals owing to the variations of individual mice in the positioning, exposure area, and imaging working distance of surrounding tissues, which was, however, not the region of interest for blood perfusion monitoring and analysis. Data are presented as mean ± SD (*n* = 5 mice). Scale bars = 2 mm. Source data are provided as a Source Data file.

oxyhemoglobin (HbO$_2$). While in the NIR-II PA images, the noise signals of Hb, HbO$_2$, lipid and water were relatively weak. Although the absorption of water increases in long-wavelength region, its absorption coefficient in the spectral region below 1350 nm (<1 cm$^{-1}$) is still much lower than that of the nanoprobe[69]. Thus its impact on thrombus PA imaging at the tested wavelengths was not pronounced. To make a better comparison, the absorption coefficients and in vivo PA signal intensities of B@SP-C NPs and main endogenous absorbers (such as water, Hb, and HbO$_2$) at various wavelengths were quantified. As depicted in Supplementary Fig. 26, Hb and HbO$_2$ showed relatively strong absorption in NIR-I region, aligning with their bright PA intensity in NIR-I region. In contrast, B@SP−C NPs manifested strong absorption and thus high PA signal in NIR-II region. Water showed

weak PA signals at all the tested wavelengths due to its low absorption coefficient. It is noted that the abundance of each absorber within tissue also greatly affects their PA signal intensity. Given the high concentration of Hb and HbO$_2$ in blood (higher than 100 g/L), the strong PA signals from Hb and HbO$_2$ under 800 nm and 950 nm light excitation were observed. In this study, the 1260 nm excitation was eventually selected for in vivo PA imaging, under which the nanoprobe showed bright PA signal while the background signal from endogenous absorbers was weak. These results also highlighted the advantage and necessity of developing highly efficient NIR-II PA probe for thrombus imaging. Under 1260 nm laser excitation, the light-up PA signal was detected in the occlusive artery after B@SP−C NPs injection, and the signal intensity reached to maximum at about 45 min post

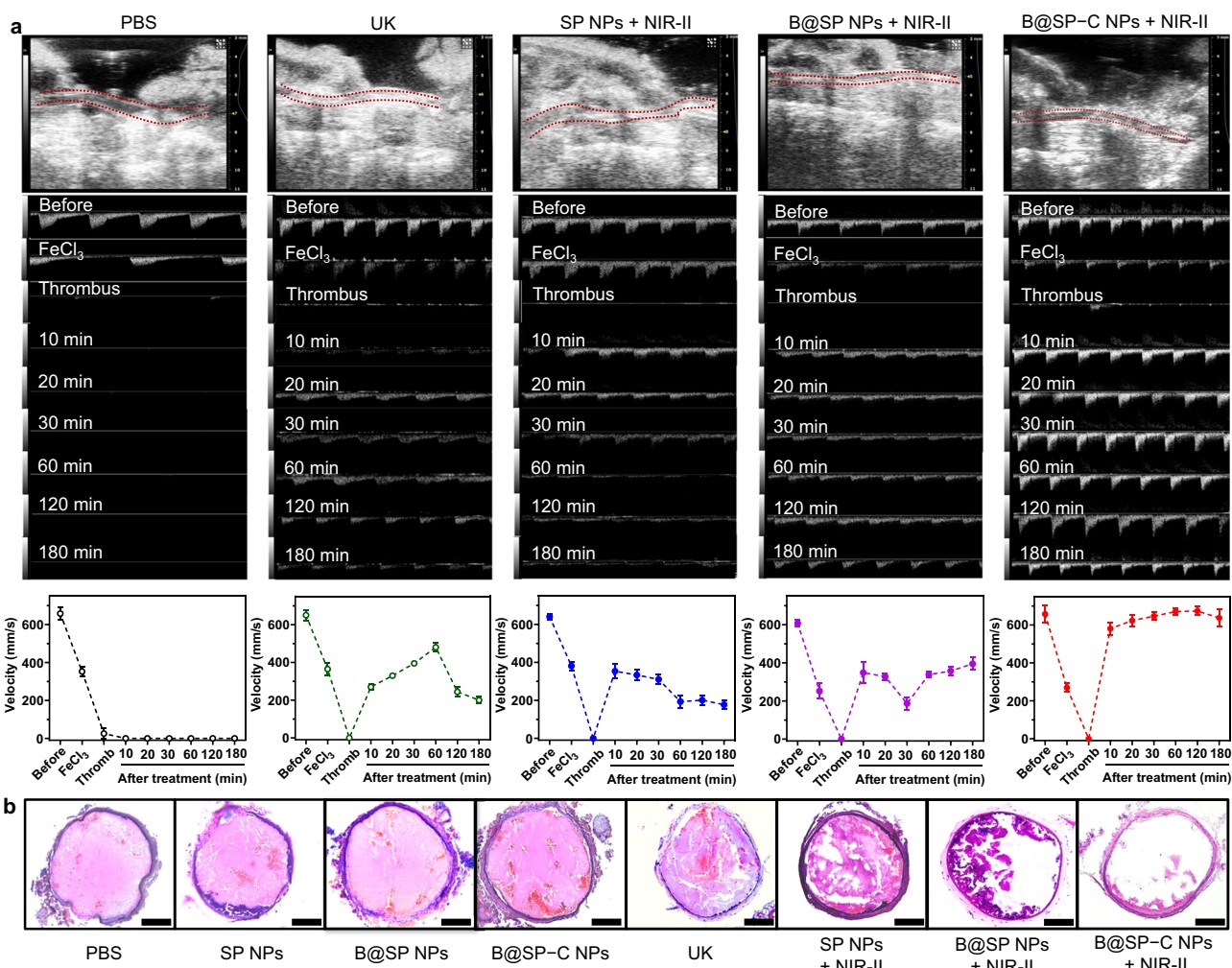

**Fig. 8 | Doppler ultrasound and histological evaluation of thrombolytic treatment. a** Representative Doppler ultrasound blood flow imaging and the corresponding blood flow velocity analysis of the mouse carotid artery after FeCl₃ induction and different therapeutic treatments, including PBS, free UK, SP NPs + NIR-II laser, B@SP NPs + NIR-II laser, and B@SP-C NPs + NIR-II laser. Data are presented as mean ± SD (*n* = 3 mice). **b** Representative H&E staining of the carotid thrombus site after different treatments (PBS, SP NPs, B@SP NPs, B@SP-C NPs, free UK, SP NPs + NIR-II laser, B@SP NPs + NIR-II laser, and B@SP-C NPs + NIR-II laser). Scale bars = 100 μm. Experiment was repeated three times independently with similar results. Source data are provided as a Source Data file.

injection (Fig. 6c, Supplementary Figs. 27, 28). As displayed in Fig. 6d, the PA intensities of B@SP − C NPs in the injured artery were around 14.1- and 3.7-folds higher than those of PBS and non-targeted B@SP NPs, respectively, suggesting excellent thrombus diagnosis capability. The PA images and SNR values of thrombus site after various treatments were displayed in Supplementary Figs. 29, 30 and B@SP-C NPs also possessed the best performance, indicating similar trend as the thrombus targeting property. The thrombotic carotid artery was also scanned at the indicated direction to acquire multiple slices of PA images at different depths (Supplementary Fig. 31). These images not only indicated the continuity of PA signals but also provided 3D tomography information about the target site. These results demonstrated that B@SP-C NPs could accurately and sensitively lighten up the thrombus in vivo, which stemmed from their excellent NIR-II PA efficiency and sufficient thrombus-targeting property.

### Photo-triggered synergistic treatment of carotid artery thrombus

Next, we investigated the potential of B@SP-C NPs as a promising nanomedicine for obstructive thrombosis management. The in vivo thrombolysis effect of different formulations in carotid artery thrombus model was evaluated according to the experimental scheme shown in Fig. 7a, and LSBFMS was applied to image and quantify the

hemodynamic changes in vasculature via live capture. First, the blood flow changes before and after the induction of carotid artery thrombus model were analyzed. After the exposure of carotid artery to 10% of FeCl₃ for 5 min, the blood flow in the model vessel was found to be dramatically retarded, and the complete and stable vascular embolism was formed after 10 min. Then different formulations, including PBS, SP NPs, B@SP NPs and B@SP-C NPs were separately administrated into the mice via tail vein. For comparation, free UK was also intravenously injected into mice at a dose of 0.04 mg kg⁻¹. Free UK was able to dissolve the clot and restored the blood flow to some extent, but the blood flow decreased again after 30 min, which was probably due to the short lift-time of macromolecular protein drugs in blood. For the NPs-injected mice, we did not observe any apparent clot lysis in the absence of NIR laser irradiation, implying that the NPs themselves had little therapeutic effect. In the laser-treated groups, the carotid arteries of mice were illuminated with 1064 nm laser (1 W cm⁻²) for 10 min at 45 min post NPs administration. First of all, the temperature change at the thrombus site under NIR-II light stimulation was monitored by a thermal infrared camera (Fig. 7b, Supplementary Fig. 32). Although these NPs had comparable photothermal conversion ability, B@SP-C NPs group demonstrated the highest temperature rise at thrombus sites due to its favorable thrombus-targeting ability. Then, we investigated whether the site-specific photothermal effect could amplify

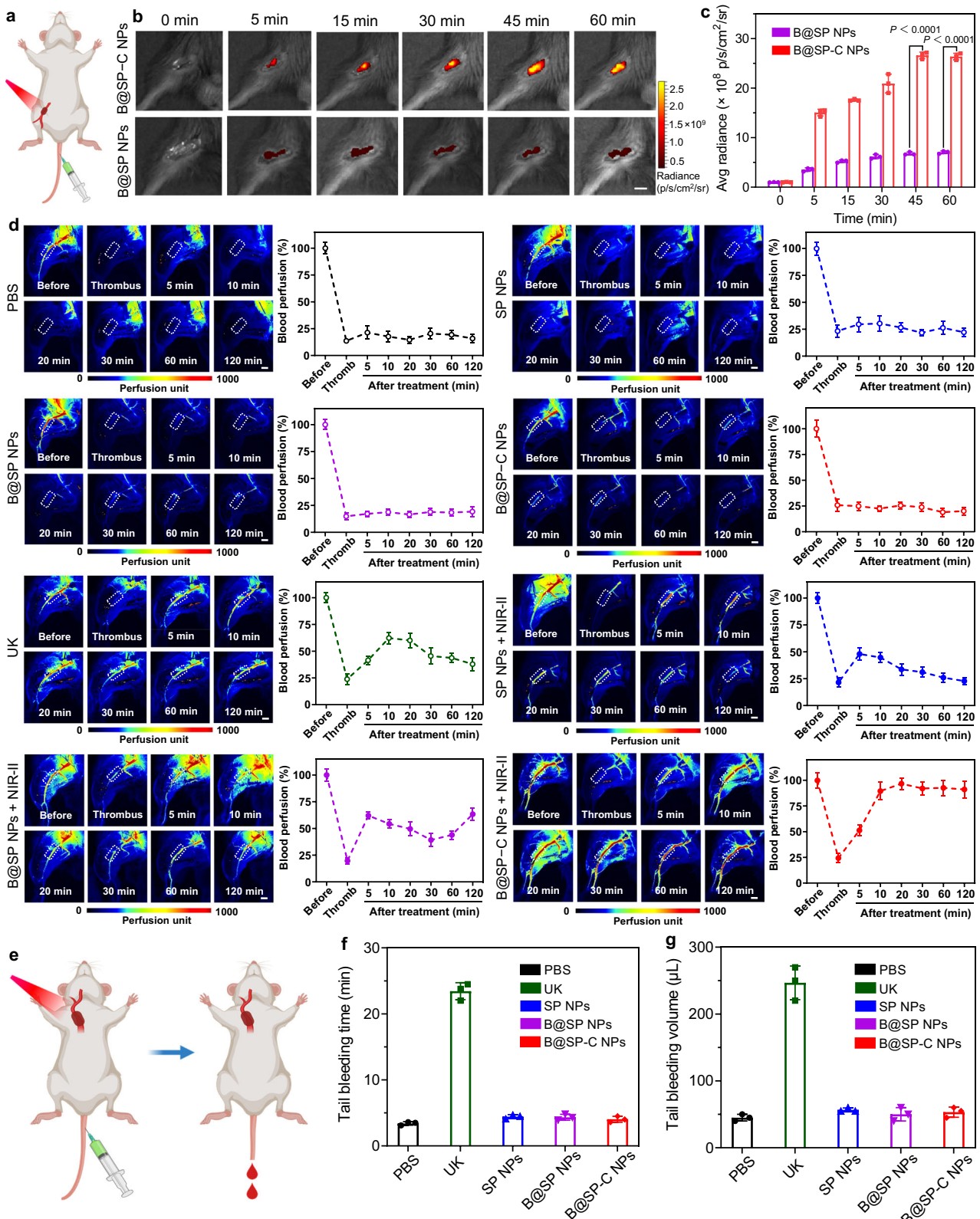

thrombolysis activity. As can be seen from Fig. 7c, SP NPs + NIR-II exposure only displayed slight antithrombotic effect, and the blood vessels were rapidly blocked again after the treatment, indicating that single photothermal thrombolysis was not potent enough for thrombus eradication and suffered from the high risk of thrombus reformation. For the BNN6-loaded B@SP NPs + NIR-II light group, more efficient thrombus removal was achieved when compared with SP NPs

+ NIR-II. The blood perfusion was gradually restored to about 68% within 2 h and there was no re-obstruction during the tested period. NO was able to reduce thrombus formation by regulating vasodilation and inhibiting platelet adhesion and aggregation, and the controlled delivery system used in this work could greatly increase the duration of NO in thrombotic lesion. In addition, the in situ NO gas generation might also promote the mechanical thrombolysis and the deep

**Fig. 9 | The evaluation of in vivo thrombolysis in the lower extremity arterial thrombosis model and hemorrhagic side effect. a** Schematic diagram of the lower extremity arterial thrombosis model. The illustration was created with BioRender.com. **b** Representative in vivo fluorescence images of the lower extremity thrombotic artery at different time points post-injection of B@SP NPs or B@SP-C NPs. Scale bar = 4 mm. **c** Quantitatively analyzing the fluorescence intensity of the lower extremity thrombotic vessels in **b**. Data are presented as mean ± SD ($n = 3$ mice). Statistical significance was calculated using two-tailed Student's *t*-test. **d** Representative LSBFMS analysis and the corresponding relative blood perfusion of the mouse lower extremity artery after FeCl₃ induction and different therapeutic

treatments, including PBS, SP NPs, B@SP NPs, B@SP-C NPs, free UK, SP NPs + NIR-II laser, B@SP NPs + NIR-II laser, and B@SP-C NPs + NIR-II laser. Data are presented as mean ± SD ($n = 5$ mice). The dotted white lines indicated the lower extremity thrombotic artery, which was selected as the region of interest to quantify the blood perfusion. Scale bars = 3 mm. **e** Schematic diagram of the tail bleeding experiment to evaluate the risk of hemorrhagic side effect. The illustration was created with BioRender.com. **f** The tail bleeding time and **g** the tail bleeding volume of mice after different treatments. Data are presented as mean ± SD ($n = 3$ mice). Source data are provided as a Source Data file.

penetration of NPs into thrombus to boost antithrombotic outcomes[68]. Thus, the integration of NIR-II photothermal thrombolysis and NO release could elicit better anti-thrombotic performance. More interestingly, owing to the excellent thrombus-homing ability, B@SP-C NPs under laser irradiation evoked a significantly improved and rapid thrombolysis, which led to about 85% recovery of blood perfusion within 20 min, and exhibited the highest thrombolysis activity among all groups.

The Doppler ultrasound blood flow imaging was also applied to track the blood flow change in thrombosis-rescue experiments (Fig. 8a). Consistent with the LSBFMS analysis, B@SP-C NPs + NIR-II light treatment again displayed the best thrombolysis capability among all the groups, which could progressively dissolve the clots and realize approximately complete recanalization of blocked artery during 20 min. After all the treatments, the carotid arteries were collected for section. H&E staining of vascular sections was then preformed to visualize the embolism in vessels (Fig. 8b). There were obvious emboli in the blood vessels of the thrombus model. The emboli size was reduced by varying degrees after different treatments. The best thrombolytic effect was induced by B@SP-C NPs plus NIR-II irradiation, as suggested by the smallest thrombotic area in artery section. The quantitative data indicated that the relative volume of clots in the B@SP-C NPs + NIR-II group was reduced by about 90% (Supplementary Fig. 33). These results proved that the thrombus-localized photothermal and NO synergistic therapy could significantly facilitate the thrombolytic performance.

In addition to carotid artery thrombosis, the lower extremity arterial thrombosis model was further developed (Fig. 9a). First, we verified that the fibrin-targeting B@SP-C NPs also displayed substantially increased accumulation to the lower extremity arterial thrombus site when compared with the non-targeted B@SP NPs (Fig. 9b, c). The therapeutic potential of different formulations on lower extremity arterial thrombosis was then evaluated by LSBFMS (Fig. 9d). As expected, the blood perfusion dramatically decreased to about 20% and maintained at a low level after the exposure of lower extremity artery to FeCl₃, revealing the successful establishment of the lower extremity arterial thrombosis model. Then the mice were randomly divided into eight groups, and treated with PBS, free UK, SP NPs, B@SP NPs, B@SP-C NPs, SP NPs + NIR-II, B@SP NPs + NIR-II, and B@SP-C NPs + NIR-II, respectively. As indicated in Fig. 9d, the NPs themselves without laser irradiation failed to produce any meaningful thrombolysis response. In the presence of light irradiation, SP NPs elicited a limited thrombolytic result with a transient and around 25% increasement of blood perfusion, and B@SP NPs improved the blood perfusion to about 63%. This indicated that the combination of NO release and photothermal thrombolysis helped to promote the blood clot removal in comparison with the only photothermal treatment (SP NPs + NIR-II), but only modestly, which may be ascribed to the inferior accumulation of B@SP NPs in thrombus sites. Notably, under NIR-II laser irradiation, B@SP-C NPs with both thrombus-binding property and multimodal therapeutic effect clearly showed the rapidest and most efficient blood clot dissolution with the blood flow almost fully recovered. These data indicated the great potential of B@SP-C NPs to efficiently reverse thromboembolism in multiple thrombus models.

## In vivo biosafety assessment

We subsequently assessed the in vivo safety profile of B@SP-C NPs. PBS or B@SP-C NPs were intravenously administrated into mice, and the major organs, including heart, liver, spleen, lung, and kidneys were collected at day 7 after injection. According to the histological images (Supplementary Fig. 34), all the organs did not show any noticeable pathological changes or inflammatory response post NPs treatment. Meanwhile, complete blood panel tests were conducted. The hematology parameters of the mice received various NPs were all within the normal ranges (Supplementary Fig. 35). Moreover, multiple serum biochemical markers for liver and renal damage, including alanine aminotransferase (ALT), aspartate aminotransferase (AST), blood urea nitrogen (BUN) and creatinine were measured for analyzing the liver and kidney functions (Supplementary Fig. 36). These hepatorenal indicators showed no significant differences between PBS and NPs-treated mice. These data collectively suggested good in vivo biocompatibility of B@SP-C NPs.

We next assessed whether the B@SP-C NPs-based thrombolysis treatment had the risk of hemorrhagic side effect. As reported previously, the conventional used thrombolytic drug, such as UK usually suffered from the unwanted bleeding complications because of the off-target action. The mice tail bleeding experiment was then carried out to evaluate the hemorrhagic side effect of different treatments (Fig. 9e). The mice were first intravenously injected with PBS, free UK, or different NPs, and then their tails were cut by a scalpel and the bleeding time of tail was monitored. For the free UK group, the tail bleeding time significantly increased (about 23.4 min), which was 6.9-folds longer than that of the PBS-injected group (about 3.4 min) (Fig. 9f). This reveled that free UK was at high risk of abnormal hemostatic complications. In contrast, all the NPs groups, regardless of whether with or without NIR-II light exposure, displayed a similar bleeding time (around 3.6−4.7 min) and bleeding volume to the PBS group (Fig. 9f, g). Lastly, we investigated the impact of NIR-II light irradiation on the blood vessel. After 1064 nm light exposure for 10 min, the treated carotid arteries of mice were collected and sectioned for H&E, elastic fiber and reticular fiber staining, and they were also evaluated by terminal deoxynucleotidyl transferase-mediated deoxyuridine triphosphate nick end labeling (TUNEL) assay. The results indicated no obvious damage to the blood vessels, including the elastic plate deformation and inflammatory reaction (Supplementary Fig. 37). Taken together, these results revealed that the localized light-activated thrombus treatment mediated by B@SP-C NPs not only presented highly potent antithrombotic efficacy but also possessed negligible hemorrhagic side effect.

## Discussion

In the present work, an advanced theranostic nanoplatform that integrated the long-wavelength NIR-II PA imaging-based precise thrombosis detection and superior antithrombotic activity was elaborately developed. A semiconducting homopolymer with strong long-wavelength NIR-II absorption and intense molecular motion that boosted photothermal and PA signal was designed and synthesized. To maximize the accuracy of thrombosis imaging and treatment, the polymer nanoprobe was decorated with a fibrin-specific ligand for

thrombus targeting. With the features of excellent NIR-II light harvesting capacity, bright PA signal, strong tissue penetration, and active thrombus accumulation ability, the NIR-II PA nanoplatform was able to sensitively and selectively delineate thrombus with an ultrahigh SNR. In addition to precise diagnosis of thrombosis, B@SP−C NPs could also be used as a photo-activated agent for non-invasive dissolution of thrombus. The combination of thrombus-localized photothermal effect and on-demand NO release not only potently dissolved blood clots but also promoted the penetration of NPs into thrombus to boost antithrombotic outcomes. Interestingly, we found that the nanoagent displayed rapid and efficient blood clot removal activity with nearly complete blood flow restoration in both FeCl$_3$-induced carotid thrombosis model and low extremity arterial thrombosis model, eliciting a significantly improved and longer-term thrombolysis response over free UK and single photothermal thrombolysis. Meanwhile, the nanoplatform was also validated to possess favorable in vivo biocompatibility and a low risk of hemorrhagic side effect, representing a safe approach for thrombolytic therapy. Although several papers have reported the detection of thrombus using PA imaging, they were based on NIR-I PA imaging[27,52]. Here we designed and synthesized a kind of high-performance NIR-II PA probe, and then demonstrated that the long-wavelength NIR-II PA imaging of thrombus could provide higher SNR and deeper tissue penetration than traditional NIR-I PA imaging. In comparison with inorganic materials, the organic polymer-based NIR-II PA contrast agent possessed the merits of relatively good biocompatibility, well-defined structure, and excellent reproducibility. Consequently, by integrating the precise NIR-II PA imaging and multimodal synergistic thrombolysis therapy into one single entity, the versatile nanoplatform developed here holds great promise to refine the current state of thrombi diagnosis and treatment.

## Methods

### Materials
DSPE-PEG and DSPE-PEG-maleimide (M.W. 2000) were purchased from Tanshtech (Shenzhen, China). Cys-Arg-Glu-Lys-Ala (CREKA) peptide (M.W. 605.7) was provided by Sangon Biotech (Shanghai, China). N,N′-Bis(1-methylpropyl)−1,4-phenylenediamine was obtained from Bide Pharmatech Ltd (Shanghai, China) and sodium nitrite (NaNO$_2$) was purchased from InnoChem Co., Ltd (Beijing, China). Iron(III) chloride (FeCl$_3$) was purchased from Heowns Biochem Co., Ltd (Tianjin, China). Thrombin (2000 U/mg) was purchased from Macklin Biochemical Co., Ltd (Shanghai, China). Cell culture dishes and confocal dishes were purchased from Corning Inocorporated Co., Ltd (Beijing, China). 3-(4,5-Dimethylthiazil-2-yl)−2,5-diphenyltetrazolium bromide (MTT) was obtained from Saiguo biotech Co., Ltd (BioFROXX, Germany). Griess reagent, reactive oxygen species assay kit (DCFH-DA), colorimetric TUNEL apoptosis assay kit and nitric oxide assay kit (DAF-FM DA) were purchased from Beyotime Biotechnology (Shanghai, China). DiR, DiI and urokinase (UK, $1.2 \times 10^5$ IU/mg) were obtained from Aladdin Biochemical Technology Co, Ltd. (Shanghai, China). Hematoxylin-eosin staining solution (alcohol soluble) and Verhoeff Elastic Fiber staining solution were purchased from Leagene Biotechnology Co., Ltd (Beijing, China). Ready-to-use dialysis bag was purchased from Ruida Henghui Technology Development Co., Ltd (Beijing, China). Centrifugal filters were obtained from Merck Millipore Co., Ltd (Merck, America). FITC-labeled fibrinogen was purchased from Qiangyao Biological Technology Co. (Suzhou, China). Reticulin stain kit (Gordon-Sweet sliver method) was obtained from Solarbio Science & Technology Co., Ltd (Beijing, China). All chemicals are analytical grade standards and require no further purification.

### Synthesis and characterization of PTIIG
2,2′-Dibromo-4,4′-bis(2-octyldodecyl)-[6,6′bithieno[3,2,b]pyrrolyli-dene]−5,5′(4H,4′H)-dione (298 mg, 0.3 mmol), hexamethyldistannane (98 mg, 0.3 mmol), Pd$_2$(dba)$_3$ (10 mg) and P(o-tol)$_3$ (30 mg) were added into a 50 mL of Schlenk tube. The mixture was degassed and charged with argon three times, followed by the addition of dry chlorobenzene (8 mL). The reaction mixture was heated to reflux and stirred for 72 h. After cooling down to room temperature, the solution was poured into 250 mL of methanol and filtered. The polymer was purified by Soxhlet extraction, and obtained as a black solid (yield 75%). GPC: $M_n$ = 32.1 kDa, $M_w$ = 55.6 kDa, PDI = 1.73. As the molecular weight of TIIG monomer is 835, the degree of polymerization was calculated to be about 38.

### Synthesis of BNN6
N,N′-Bis(1-methylpropyl)−1,4-phenylenediamine (BPA, 1.17 mL) was diluted into ethanol (10 mL) and then mixed with NaNO$_2$ solution (6 M, 10 mL) under argon protection with stirring. After 30 min, the aqueous solution of HCl (6 M, 10 mL) was added dropwise. After stirring for 4 h, the pale-yellow product was collected by centrifugation at $300 \times g$ for 15 min. The precipitate was washed with ethanol repeatedly and then dried under a freezing vacuum to afford N,N′-di-sec-butyl-N,N′-dinitroso-1,4-phenylenediamine (BNN6) as a colorless solid.

### Synthesis of DSPE-PEG-CREKA
CREKA was reacted with DSPE-PEG-maleimide via thiol-Michael addition reaction in water/methanol solution (90:10, v/v) for 8 h at room temperature. Then the residual CREKA peptide was removed by centrifugal filters (Merck Millipore Co., Ltd, Shanghai China) and the final product DSPE-PEG-CREKA was obtained by lyophilizing.

### Preparation of NPs
The nanoformulations were prepared by a modified nanoprecipitation method. The synthesis process was conducted at the temperature of 20 °C. First, PTIIG (1 mg), DSPE-PEG (4 mg), and DSPE-PEG-CREKA (4 mg) were dispersed in 0.5 mL of THF and then mixed with BNN6 solution (0.5 mL, 2 mg mL$^{-1}$) under stirring. After the polymer was completely dissolved, the above stock solution was added dropwise into 10 mL of purified water under vigorous stirring. The organic solvent was evaporated by stirring in a fume hood, and then the NPs were concentrated through the ultrafiltration filter (Millipore). The resulting B@SP−C NPs were stored in a 4 °C refrigerator for subsequent experiments. The non-targeted B@SP NPs were prepared by the same method, except that DSPE-PEG-CREKA was replaced with DSPE-PEG.

### Characterization of NPs
The mean hydrodynamic diameter and zeta potential of SP NPs, B@SP NPs, and B@SP−C NPs were determined by DLS (Malvern Zetasizer Nano-ZS, UK) in phosphate buffer (0.01 M and pH = 7.4) at 25 °C. The morphology of the as-synthesized NPs including SP NPs, B@SP NPs, and B@SP−C NPs were examined by field emission TEM (FEI, Talos L120C G2, Czech Republic). FT-IR (TENSOR 37, Germany) was used to confirm the composition of the nanoformulations. The absorption spectra were obtained by UV-vis-NIR spectrophotometer (UV-3600 plus, Shimadzu Manufacturing Company, Japan). Fluorescence spectrum and ROS generation were detected by FLS1000 (Edinburgh, UK). The loading amount of BNN6 in B@SP−C NPs was measured according to the standard curve of BNN6. For this, standard solutions of BNN6 at different concentrations (0−1 mg mL$^{-1}$) were prepared, and their UV−vis absorbance spectra were measured. The standard curve of absorbance at 375 nm versus BNN6 concentrations was plotted, which was utilized to calculate the loading amount of BNN6 in B@SP−C NPs.

### Photothermal property of B@SP−C NPs
To investigate the photothermal performance of B@SP−C NPs, different concentrations of B@SP−C NPs aqueous solution (0, 6.25, 12.5,

25, 50, 100, and 150 μg mL⁻¹ based on PTIIG) were respectively exposed to 1064 nm (1 W cm⁻²) laser (Changchun Laser Optical Technology Co., Ltd., Changchun, China) for 10 min. The temperature profiles and images were precisely recorded with a thermal imaging camera from FLUKE technology. In addition, different laser power intensities (0.25, 0.5, 0.75, 1.0, 1.25, and 1.5 W cm⁻²) were employed to investigate the photothermal effect of NPs. In order to study its photostability, five cycles of laser on/off were carried out. First of all, B@SP−C NPs were dissolved in PBS (2 mL) and placed in a quartz colorimetric tube (5.8 g), and then irradiated with a 1064 nm laser (1 W cm⁻²) for 10 min to reach thermal equilibrium and then naturally cooled. The calculation process is as follows:

The energy input and dissipation of the measurement system can be expressed as:

$$\sum_i m_i\, C_{p,i}\, \frac{dT}{dt} = Q_{NP} + Q_{sys} - Q_{diss} \tag{1}$$

where $m_i$ and $C_{p,i}$ are the mass and heat capacity of the component (e.g., water and quartz cuvette). $Q_{NP}$ is the energy input from B@SP−C NPs, $Q_{sys}$ is the energy input from the other components of the system, and $Q_{diss}$ is the energy loss/dissipation energy from the system to the surroundings.

$Q_{NP}$ could be expressed as:

$$Q_{NP} = I(1 - 10^{-A\lambda}) \tag{2}$$

where $I$ is the power of 1064 nm laser, $A\lambda$ is the absorbance of B@SP−C NPs at 1064 nm;

Instead of nanoparticles, pure solvents can be used to measure $Q_{sys}$ and denoted as:

$$Q_{sys} = hS(T_{max,solvent} - T_{surr}) \tag{3}$$

$Q_{diss}$ could be expressed as:

$$Q_{diss} = hS(T - T_{surr}) \tag{4}$$

where $h$ is the heat transfer coefficient; $S$ is the surface area of the quartz cuvette exposed to laser; $T$ is the temperature at a predetermined point in time; $T_{surr}$ is the ambient temperature. $T_{max}$ is the temperature recorded when the measurement system reached thermal equilibrium.

At this time, the total energy input to the system is equal to the energy dissipated.

$$Q_{NP} + Q_{sys} = Q_{diss} = hS(T_{max} - T_{surr}) \tag{5}$$

After removal of laser irradiation, the energy input drops to zero, and Eq. (1) is expressed as:

$$\sum_i m_i C_{p,i} \frac{dT}{dt} = -Q_{diss} \tag{6}$$

Equations (4) and (6) can give:

$$t = -\frac{\sum_i m_i C_{p,i}}{hS} ln\frac{T - T_{surr}}{T_{max} - T_{surr}} \tag{7}$$

$\frac{\sum_i m_i C_{p,i}}{hS}$ was defined as $\tau_s$, and $\frac{T-T_{surr}}{T_{max}-T_{surr}}$ was defined as $\theta$.

Equation (7) can be expressed as

$$t = -\tau_s ln\theta \tag{8}$$

where $\tau_s$ could be obtained from a linear regression of time on negative $ln\theta$.

Finally, $\eta$ (PCE) can be calculated as:

$$\eta = \frac{hS(T_{Max}T\,max - T_{Surr}) - Q_{diss}}{I(1 - 10^{-A\lambda})} \tag{9}$$

## Measurement of NO generation
In order to evaluate the NO generation capacity, Griess assay kit was used to quantitatively measure the NO concentration under 1064 nm laser irradiation. Typically, different concentrations of BNN6, SP NPs, B@SP NPs and B@SP−C NPs solutions were irradiated by a 1064 nm laser (1 W cm⁻²) for different time. After irradiation, the NPs were precipitated and separated by centrifugation. The supernatant was incubated with the Griess reagent for 20 min at 37 °C. The nitrite concentration was analyzed by measuring the optical density at 540 nm (OD540 nm) using a microplate reader (spark, Austria). The NO concentration was calculated using the calibration curves based on different concentrations of nitrite ion solutions.

## Cells and animals
The mouse embryonic fibroblasts line (NIH 3T3) was purchased from Chinese Academy of Sciences Cells Bank (Shanghai, China) and the human umbilical vein endothelial cell line (HUVEC) was purchased from Cyagen Biosciences (Guangzhou, China) Inc. All cells were cultivated in a humidified atmosphere at 37 °C with 5% of CO₂.

Male C57BL/6 mice (6-week old) were purchased from the Laboratory Animal Center of the Academy of Military Medical Sciences (Beijing, China). The living environment of animals were maintained at a temperature of 25 °C with a 12 h light/dark cycle, suitable humidity (typically 50 %) with free access to standard food and water. The mice were randomly selected from the cage to assign into different experimental groups. All procedures involving animal were conducted in accordance with the guidelines set by the Tianjin Committee of Use and Care of Laboratory Animals, and approved by the Animal Ethics Committee of Nankai University (2021-SYDWLL-000406).

## Intracellular NO measurement
The NO release from the B@SP−C NPs-treated HUVECs was measured using the commercial NO fluorescent probe, DAF-FM DA. DAF-FM DA can cross the plasma membrane and be cleaved by esterases to generate intracellular DAF-FM, which is then oxidized by NO to a triazole product with increased fluorescence. Typically, HUVECs were cultured in complete EGM-2 medium at 37 °C in a CO₂ incubator. The HUVECs were seeded in confocal cell culture dishes. After 24 h, the cells were incubated with B@SP−C NPs (0.1 mg mL⁻¹ based on PTIIG) for 12 h. Then cells were washed three times with PBS buffer. As followed, the NO probe DAF-FM DA (the final concentration was 5 μM) was added and incubated at 37 °C for 30 min, and washed three times with PBS. The cells were irradiated with a 1064 nm laser (1 W cm⁻²) for 5 min to trigger the NO release. Finally, the generation of NO from B@SP−C NPs-treated HUVECs was detected by imaging cells using confocal fluorescence microscopy (LSM 800 with Airyscan, ZEISS, Germany) at excitation and emission wavelengths of 495 and 515 nm, respectively. Statistical analysis and graphing were done with ZEN 2012 (Version 1.1.2.0).

## Cytotoxicity of B@SP−C NPs
MTT assay was used to evaluate the cytotoxicity of B@SP−C NPs. HUVECs and NIH 3T3 cells were seeded in a 96-well plate for 12 h to reach 80−90% confluency. Cells were treated with various

concentrations of B@SP–C NPs for 24 h. 10 μL of MTT solution was added into each well and incubated for 4 h. Dimethyl sulfoxide (150 μL) was added to dissolve the resulting formazan crystals. After that, the absorbance at 570 nm was measured on the multifunction enzyme marker (spark, Austria), and the absorbance of each experimental group was compared with the control group to determine the cellular viability.

## In vitro hemolysis experiment

The in vitro hemolysis experiment is an important way to evaluate the blood compatibility of NPs. For the hemolysis evaluation, fresh blood was collected from C57BL/6 mice (6 weeks old) with ethylenediaminetetraacetic acid (EDTA)-containing tube, and the whole blood was centrifuged ($380 \times g$), washed, and diluted ten times with PBS to obtain clean red blood cell suspensions. Different NPs solutions (200 μL) were added to 1 mL of red blood cell suspensions and incubated for 4 h at room temperature. Red blood cell suspensions mixed with water were used as the positive control, and red blood cell suspensions mixed with PBS were used as the negative control. Subsequently, the supernatant was collected by centrifugation ($13,500 \times g$) and the absorbance at 540 nm was measured using a microplate reader (spark, Austria). The hemolysis percentage is calculated as follows: hemolysis (%) = (sample group absorbance – PBS group absorbance)/(water group absorbance – PBS group absorbance) × 100%.

## In vitro thrombus targeting of B@SP NPs and B@SP–C NPs

Fresh blood was collected and divided into tubes with equal volumes (50 μL). Each tube was mixed with thrombin (5 U μL$^{-1}$) and CaCl$_2$ (3 mM) to induce the formation of clots. The artificial thrombus was then respectively incubated with PBS, DiR-loaded B@SP NPs or DiR-loaded B@SP–C NPs aqueous solution (0.1 mg mL$^{-1}$) to verify the targeting ability of NPs. After incubation for 1 h, 3 h or 5 h, the thrombus clots were taken out and washed three times with PBS. The fluorescence on thrombus clots was analyzed using the small animal imaging system (PerkinElmer, IVIS Spectrum).

## In vitro thrombolytic efficacy

The artificial thrombus was placed into the 5 mL of glass vial, to which the mixture of 4.5 mL of PBS and 0.5 mL of different NPs solutions (SP NPs, B@SP NPs, or B@SP–C NPs, 1 mg mL$^{-1}$ based on PTIIG) was added. The mixture was irradiated with a 1064 nm laser for 20 min and then incubated at 37 °C. The weights of thrombus before and after thrombolytic treatment were measured to calculate the thrombolysis rate, and thrombolysis rate = (weight before treatment – weight after treatment)/weight before treatment. In addition, the treated thrombus clots were sectioned and stained with hematoxylin and eosin (H&E) to evaluate the thrombolysis efficiency.

## In vitro PA properties of B@SP–C NPs

The PA imaging was carried out on an ultra-high resolution ultrasound/photoacoustic multimode imaging system for small animals (Vevo® LAZR-X, Fuji VisualSonics, Canada). For evaluating the PA performance of B@SP–C NPs in vitro, different concentrations of B@SP–C NPs solution were imaged under Vevo® LAZR-X system. In addition, the PA signals of B@SP–C NPs solution covered by different thicknesses of chicken breast were also measured according to previous paper[70]. The setting parameters for PA imaging were as follows: frequency = 30 MHz, wavelength range = 680–970 nm/1200–2000 nm (10 mJ cm$^{-2}$), PA gain = 33 dB, depth/width = 11.00/15.36 mm, pulse repetition rate = 20 Hz, duration of pulsed laser = 7 ns. It was noted that such short-duration and long-break excitation light can hardly induce noticeable increase in the system temperature. In addition, since it is necessary to have medium to transmit the acoustic waves, both the transducer and the test specimen were immersed in water during the in vitro PA measurement process, which further helped to maintain a stable temperature. Thus, the influence of temperature during PA measurement was negligible.

## Construction of thrombus model

The male C57BL/6 mice (6-week old) were anesthetized using isoflurane and the neck hair was shaved off. Then the skin around the neck was cut with surgical scissors, and the connective tissue and fat were peeled away to expose the carotid vessels. A 10% of FeCl$_3$-soaked filter paper (3 × 1 mm) was placed on the surface of the exposed carotid artery for 5 min, then the filter paper was removed. The vessel and surrounding tissue were washed with sterilized saline solution. After ~10 min, an apparent aggregated embolus could be observed under the somatic microscope. The laser speckle imaging system (RWD, RFLSI III, Shenzhen, China) was used to monitor the hemodynamic changes before and after the induction of carotid artery thrombus using FeCl$_3$. The mice lower limb arterial thrombosis model was also established using the similar protocol as described above. Briefly, the leg skin of mice was excised and the fat or other connective tissues were stripped away to expose the arteries in the lower limb. Filter paper infiltrated with 10% of FeCl$_3$ solution was taped to the surface of the lower extremity arteries for 5 min, then removed and washed with saline. The blood flow changes before and after thrombus induction was also measured by using the laser speckle imaging system.

## Pharmacokinetic analysis, tissue distribution and thrombus targeting of B@SP–C NPs in vivo

The male C57BL/6 mice (6 weeks old) were intravenously injected with DiR-loaded B@SP–C NPs (10 mg kg$^{-1}$). At different time intervals (0, 1, 3, 6, 12, 24, 36, 48, and 72 h) after injection, the blood was collected from the tail artery and the fluorescence intensity of blood samples was analyzed using a microplate reader at excitation and emission wavelengths of 750 and 780 nm respectively to determine the plasma concentration of NPs. The clearance of free UK in blood was measured by UK assay kit. The obtained fresh blood was immediately centrifuged to get the supernatant serum, which was then analyzed with the UK assay kit (Urokinase Activity Assay Kit, Abcam) according to the manufacturer's instructions. To study the metabolism process of NPs, the feces and urine of mice injected with the DiR-loaded B@SP–C NPs were collected at different time points post NP administration. Then the NP fluorescence signals in feces and urine were monitored using the IVIS system (PerkinElmer).

To investigate the biodistribution of nanoformulations in vivo, DiR-B@SP NPs or DiR-B@SP–C NPs (10 mg kg$^{-1}$ based on PTIIG) were systematically administered to the thrombotic animals via tail vein. The mice were imaged under in vivo imaging system (IVIS, PerkinElmer, America) at different time points (0, 5, 15, 30, 45, and 60 min) post NPs injection. In addition, the mice were sacrificed at 2 h after NPs injection, and the major organs including heart, liver, spleen, lung, kidneys as well as the carotid artery or lower extremity arteries were collected for ex vivo fluorescence imaging using IVIS. The fluorescence intensity of different tissues was quantified and compared. To investigate the thrombus targeting mechanism of B@SP–C NPs, free CREKA was administered 5 min prior to B@SP–C NPs injection to occupy the fibrin binding sites, and then the accumulation of NPs to the free CREKA-pretreated thrombus site was analyzed using the same procedure as mentioned above. The concentrations of B@SP–C NPs in the region of thrombosis were also quantified. The DiR-loaded B@SP–C NPs were injected into the carotid thrombosis-bearing mice, and then both the thrombotic carotid artery and the contralateral healthy carotid artery were collected 1 h post NP administration. The vessel tissues were weighted and cut into small pieces, and then homogenized using a Bioprep-24 Homogenizer. The DiR in the tissue homogenate was extracted by methanol, and a standard curve between the fluorescence intensity and DiR concentration was built to calculate the concentration of DiR-loaded B@SP–C NPs in blood vessels. The

fluorescence of DiR was recorded with a microplate reader (Spark, Austria, Ex: 741 nm; Em: 786 nm).

## In vivo NIR-II PA imaging of thrombus

The C57BL/6 mice (6 weeks old) with thrombus model were intravenously administered with B@SP NPs or B@SP−C NPs (10 mg kg$^{-1}$ based on PTIIG), respectively. And the mice were then anesthetized using isoflurane (2% of isoflurane, 0.5 mL min$^{-1}$ oxygen), and placed on a heating platform at 37 °C. At predetermined time points (0, 5, 15, 30, 45, 60, and 90 min) post NPs administration, the in vivo NIR-II PA imaging was carried out using Vevo® LAZR-X system. The setting parameters for PA imaging system were as follows: frequency = 30 MHz; wavelength range = 680–970 nm and 1200–2000 nm; PA gain = 33 dB; gain = 13 dB depth/width = 11.00/15.36 mm; wavelength = 800 nm, 950 nm, 1260 nm and 1350 nm; scanning area: ~165 mm$^2$. Simultaneous output of ultrasound images and photoacoustic images were acquired to ensure the positioning of the photoacoustic signal. The merged PA/US images were obtained by superimposing each PA image in pseudo-color on the gray-scale US image. The pseudo-colored PA images were reconstructed by Vevo LAB 5.6.1 software, and the intensity of the pseudo-color in each PA image reflected the intensity of local PA signals. The spectral unmixing was carried out using the commercial software (Vevo LAB 5.6.1) provided by the Vevo® LAZR-X imaging system (Fuji VisualSonics, Canada). The spectral unmixing is based on the PA spectral signatures of each absorber within the sample, and the spectral unmixing algorithm used in this software has been reported in the literature[71]. Currently, several spectral unmixing methods and algorithms have been developed to identify different constituents. However, the accuracy of spectral unmixing is still affected by the set of wavelengths selected, local fluence variations, the high complexity of biological tissues, and the overlap of spectral features among different absorbers[72,73]. In further work, we will try to optimize the spectral unmixing methods to achieve better spectral unmixing analysis outcomes.

## Evaluation of the thrombolytic effect of different formulations in vivo

To evaluate the therapeutic effect of different formulations on thrombi, the mice with carotid thrombosis or the lower extremity arterial thrombosis were randomly divided into eight groups ($n = 5$ mice): PBS, SP NPs, B@SP NPs, B@SP−C NPs, UK, SP NPs + NIR-II, B@SP NPs + NIR-II, B@SP−C NPs + NIR-II. And different formulations were intravenously injected into mice via tail vein. For the NIR light-treated groups, the carotid arteries or lower extremity arteries of mice were illuminated with 1064 nm laser (1 W cm$^{-2}$) for 10 min at 45 min post NP administration. The temperature changes at the irradiation sites were monitored using an infrared thermal imaging system (FLK-Ti200, Shanghai, China). At different time points (5, 10, 20, 30, 60, and 120 min) after the treatments, the laser speckle imaging system was used to observe and record the blood flow velocity and blood flow at thrombosis sites. In addition, the Doppler ultrasound imaging system (VEVO, Vevo2100, America) was also applied to measure the blood flow velocity at different time points post treatments. Finally, the mice were euthanized, and the carotid arteries or lower extremity arteries were excised for section and H&E staining. The areas and morphology of the thrombi after different treatments were observed under optical microscope. The sections were analyzed using the Image J2x software (version 2.1.4.7) by an investigator blinded to the treatments, and thrombolytic efficiency was determined by the area ratio of vascular occlusion to total vascular lumen.

## In vivo biosafety evaluation

To test the potential toxicity of B@SP−C NPs, healthy C57BL/6 mice (6-week old) were intravenously injected with PBS or B@SP−C NPs (10 mg kg$^{-1}$ based on PTIIG). One week after PBS or NPs administration, the mice were euthanized, and blood was drawn. Then the hematological parameters, including white blood cell count, red blood cell count, hemoglobin concentration, hematocrit, mean red blood cell volume, platelet count, mean corpuscular hemoglobin content, and mean corpuscular hemoglobin concentration were analyzed using an automated hematology analyzer. And the serum was collected for evaluating multiple blood biochemical parameters, including alanine aminotransferase (ALT), aspartate aminotransferase (AST), blood urea nitrogen (BUN) and creatinine. The major organs (heart, liver, spleen, lung, and kidneys) were also harvested for histological analysis. Briefly, the tissues were fixed in 4% of paraformaldehyde (PFA), followed by embedding in paraffin and sectioning at a thickness of 5 μm for H&E staining. The slices were observed under a digital microscope (Leica QWin). Finally, the hemorrhagic side effects of different treatments were evaluated by tail bleeding experiment. The mice were first intravenously administrated with PBS, free UK or different NPs, and then treated with NIR-II laser using the same protocol as for thrombolytic therapy. And then the mice were anesthetized, and their tails were amputated with a scalpel. The tail was placed in a tube, and the bleeding time and volume of tail were monitored. Bleeding time was defined as the time required for the wound to stop bleeding for at least 10 s. Lastly, the impact of NIR-II light irradiation on the blood vessel was investigated. After 1064 nm light (1 W cm$^{-2}$) exposure for 10 min, the treated carotid arteries of mice were collected and sectioned for H&E, elastic fiber and reticular fiber staining, and they were also evaluated by terminal deoxynucleotidyl transferase-mediated deoxyuridine triphosphate nick end labeling (TUNEL) assay according to the manufacturer's instructions.

## Statistical analysis

Statistical analysis and graphing were done with GraphPad Prism 8.0.2. All data were expressed as the mean value ± standard deviation. Statistical comparisons were made by two-tailed Student's $t$-test (between two groups) and one-way ANOVA (for multiple comparisons). $P$ value < 0.05 was considered statistically significant.

## Reporting summary

Further information on research design is available in the Nature Portfolio Reporting Summary linked to this article.

## Data availability

All data supporting the findings of this study are available within the article and its supplementary files. Supplementary Dataset 1 presents the Cartesian coordinates of oligomers with different numbers of repeat units (TIIG1−5) by DFT calculation. Any additional requests for information can be directed to, and will be fulfilled by, the corresponding authors. Source data are provided with this paper.

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

## Acknowledgements

This work was financially supported by the NSFC (81921004 (D.K.), 82172081 (J.Q.), 82102200 (W.L.), and 52103168 (J.Q.)), CAMS Initiative for Innovative Medicine (2021-I2M-1-043 (W.L.)), the Science and Technology Program of Tianjin, China (21JCZDJC00970 (J.Q.) and 22JCYBJC01000 (W.L.)), and the Fundamental Research Funds for the Central Universities (63233052 (J.Q.), 63231199 (J.Q.), and 2021-RC350-006 (W.L.)).

## Author contributions

J.S., D.K., W.L., and J.Q. conceived and designed the research. J.S., X.K., and L.W. performed the research. J.S., D.D., W.L., and J.Q. analyzed the data and participated in the discussion. J.S., W.L., and J.Q. contributed to the writing of this paper. D.K., W.L., and J.Q. obtained funding for data collection and analysis, supervised the study and provided strategic guidance.

## Competing interests

The authors declare no competing interests.
