## [Peer review file · Nature Communications]

REVIEWER COMMENTS

Reviewer #1 (Remarks to the Author):

The manuscript entitled "Near-infrared-II photoacoustic imaging and photo-triggered synergistic treatment of thrombosis via fibrin-specific homopolymer nanoparticles", reports a phototherapy-based method for the thrombolysis. In this design, Song et al. synthesized a targeting peptide modified DSPE-PEG-based liposome to load the NIR-II-light responsive polymer (PTIIG) and the BNN6 as the Nitric Oxide donor. The characterization of the nanosystem demonstrated that the NPs has strong absorbance in second near infrared window and the adsorbed energy can induce the local temperature rise and the acoustic wave for the photothermal therapy and photoacoustic imaging. The ex vivo and in vitro experiments demonstrate that with the modified targeting peptide, the NPs can accumulate more in the thrombosis lesion and under the NIR-II laser, and that the NPs can be used as PA probe and thrombolytic tool. The in vivo experiments further demonstrate the NPs synergistic capacity for the diagnosis and treatment of thrombosis.

Although a very interesting study and well supported by the data, the major issue is the NIR-II strategy is not very novel. Authors should demonstrate in introduction the advantages of their nanosystem compared to several of those already reported in the literature, as well as the potential clinical translation.

Furthermore, other major issues are indicated below.

1. In Figure 3 a, the observed size of the NPs in the TEM is not inline with the DLS results and the claimed 126 nm in the text. This needs to be further clarified.
2. For the NPs fabrication, authors should demonstrate explain how to make sure the encapsulation of BNN6 and SP NPs is uniform through the self-assembly of DSPE-PEG.
3. For the photothermal therapy, 1064 nm laser with 1W/cm² was used for 10 minutes' laser irradiation. With this high power, could it be harmful for the local treatment area? Because for the thrombosis laser treatment, the vessel rupture, hemorrhage and the purpura can occur, the tail bleeding experiment cannot show the hemorrhagic side effect of the local treatment area. Further experimental demonstrations should be provided here.
4. Authors should provide more TEM pictures of the NPs (SP NPs, B@SP NPs, and B@SP-C NPs) in lower magnification with more particles for the uniformity determination.
5. In the text, the order of Figure S 9 and Figure S 10 is reversed.
6. Authors should provide the FTIR spectrum of free CREKA, otherwise, it cannot be claimed in the text that "the characteristic bands of BNN6 and CREKA peptide appeared in the FTIR spectrum of B@SP-C NPs" since the spectra in the Figure S 11 are similar.
7. Authors should explain how to evaluate clearance of the free injected UK as comparison, based on blood test or fluorescence.
8. Since the synthesized SP NPs and B@SP NPs are hydrophobic, Authors should provide more information about how to disperse those NPs in the injection solvent.
9. Finally, another major issue is the rather low number of animals used for each of the indicated animal experiences.

Reviewer #2 (Remarks to the Author):

The authors show a very nice and concise story for a theranostic imaging of thrombi using PA imaging. Further comments primarily apply to the PA imaging. While the PA study is nicely done and seems convincing to me I would appreciate a number of additions and controls:

- 1) Other works showing thrombus detection with NPs in PA (for example: 10.2147/IJN.S216603) should be discussed. And comparisons on the performance would be worthwhile.
- 2) Why is the different absorption (3f) in no way reflected in the PA signals (4f). Given the difference

in absorbance the PA signal should at least be slightly different for the 3 different NPs, no?

3) Please add info on how the anatomy BG images in 7a and c are obtained. And how the false color? Is the anatomy always the same WL? If so, why are they looking slightly different in the different WL. If at different WL how is the false color generated – just the raw signal at the indicated WL?

4) Signals outside artery not present in PBS, where do they come from? E.g. 7c top row 30min.

5) It would be nice to show multiple slices (in the suppl) of the in vivo PA images to show continuity of the artery PA signal.

6) Please show all mice data as similar scaled images in the supplement. The error bars (7d) for a set of in vivo measurements I find impressively small.

7) With a tunable laser source as you have in the LAZR-X system you could scan around the peak of your agent and even improve detectability by spectral unmixing. It would be nice when this could be added to show the potential of the method.

Reviewer #3 (Remarks to the Author):

This manuscript seeks to demonstrate a polymer nanopatform that integrates the long-wavelength near infrared-II (NIR-II) photoacoustic (PA) imaging with nanoparticles delivery system. This approach is interesting to address the unmet clinical need for detecting thrombus formation and to provide anti-thrombotic activity in the cardiovascular space. The authors demonstrated a semiconducting homopolymer nanoparticles with fibrin-specific peptide for NIR-II absorbance, photo-to-heart conversion, and selective thrombus targeting. However, the performance of the nanoparticle would be strengthened with data to support selectivity, potency, and biocompatibility. The following are major and minor comments to address the results and conclusion.

Major Comments:

1. Unclear are how to accumulate the nanoparticles near the thrombosis, and how to demonstrate the selectivity of fibrin-specific peptide. Please provide the concentration of nanoparticles near the regions of thrombosis as compared to the non-thrombosed blood vessels.

2. Please address how the nanoparticles are degraded and secreted from the circulation.

3. Please provide the duration required to remove the thrombus formation and the percentage of thrombus that can be removed using the on-demand NO release. Using NO to remove thrombosis is recognized to require a long duration, whereas the half-life of NO is short. Please demonstrate the thrombus removal efficiency by NO in response to FeCl₃-induced carotid thrombosis and lower extremity arterial thrombosis models.

5. In Figure 3, dynamic light scattering, and transmission electron microscope should have different results to measure the average diameter of nanoparticles.

6. In Figure 4b, please follow a standard protocol to measure PA signal of B@SP-C NPs and provide the references. Figure 4b shows plateaus after 10 mm penetration depth, whereas light attenuation at different penetration depths is exponential. The plateaus could suggest the presence of stray light or other sources of illumination that bypassed the thickness. Thus, the authors should double-check the experiment settings and investigate the sources of the observed plateau. In Figure 4h, how long could B@SP-C NPs endure under laser irradiation? Please enlarge the time of laser irradiation.

7. In Fig. 6a, the B@SP-C NPs exhibited an improved half-life. However, the graph also indicates that

after 25 hours, approximately 30% of the injection remained unclear. Therefore, it is essential to evaluate whether the residual of the NPs after a prolonged period could pose a concern.

8. The legend of Fig. 7 indicates that the images shown are NIR-II PA images. However, it is unclear whether these are purely PA images or a combination of PA and ultrasound images. The significance of the two different color bars used was not explained.

9. The selected field of view in some of the images was too small for adequate comparison. Therefore, the authors may consider including larger or closer images of the region of interest to enable better comparison.

10. In Fig. 7a, the 800 nm and 950 nm wavelengths showed very strong surface signals, whereas the 1260 nm and 1350 nm wavelengths did not show any. The cause of the strong surface signals should be explained. Moreover, comparing SNR across these wavelengths can be challenging due to differences in water absorption, and the authors may need to confirm that the same laser influence after water is present across all wavelengths.

11. In Fig. 8c, the authors should clarify how the blood perfusion was quantified for the entire image or a selected field of view. It is unclear whether some of the figures demonstrate an increase in flow, whereas the corresponding plot shows otherwise. Finally, the color bar used only shows high vs. low. Please provide a quantitative scale that correlates to specific flow numbers.

Minor issues:

1. In the third paragraph of the introduction, please provide the penetration depth ranges of different optical technology in the human body.

2. In the Introduction, the authors stated that NIR-II light-excited PA imaging could provide higher resolution and signal-to-noise ratio (SNR), as well as deeper tissue penetration due to reduced light scattering and attenuation. However, the resolution of photoacoustic computed tomography is not dependent on the wavelength used. Additionally, the NIR-II light is known to be influenced by water and lipid absorption, which may not improve the SNR as compared to NIR-I.

3. Please provide biomedical references to support the statement that "there is growing evidence showing that localized hyperthermia could accelerate the ablation of blood clot via loosening the non-covalent interactions of fibrins, providing a safe and non-invasive thrombolytic approach". The current references were mostly in chemistry and materials science.

4. Please provide the specific temperature at which the nanoparticles were synthesized.

5. Line 112: There seems to be a typo: "Photo-to-heart" vs. "Photo-to-heat".

6. Fig.1: Please elaborate in the caption of Fig.1 to help understand the schematics.

7. Please provide a scale bar on the fluorescence intensity in Fig. 6e.

8. Figures 5d and 6e lack the scale bars.

9. The color bars in the figure are not clearly illustrated. For example, in figures 5b, 6c, 8c, please quantify "high" vs. "low" in the figures or legends.

10. This manuscript contains 10 Figures. Some Figures may be considered as supplementary information, such as Figure 2. Please address the units in the synthetic PTIIG (Figure 2)?

Responses to reviewers' comments for the manuscript titled "Near-infrared-II photoacoustic imaging and photo-triggered synergistic treatment of thrombosis via fibrin-specific homopolymer nanoparticles".

We sincerely thank the reviewers for their positive feedback and valuable comments concerning our article. These comments and suggestions are all of great importance to improve the quality of our article. The point-by-point responses to those comments are presented as below and all changes to the manuscript are highlighted by using red colored text. We hope that the reviewers are satisfied with these changes and the manuscript will be accepted for publication in Nature Communications.

Responses to the Comments and Suggestions of Reviewer #1

Reviewer #1 (Remarks to the Author):

The manuscript entitled "Near-infrared-II photoacoustic imaging and photo-triggered synergistic treatment of thrombosis via fibrin-specific homopolymer nanoparticles", reports a phototherapy-based method for the thrombolysis. In this design, Song et al. synthesized a targeting peptide modified DSPE-PEG-based liposome to load the NIR-II-light responsive polymer (PTIIG) and the BNN6 as the Nitric Oxide donor. The characterization of the nanosystem demonstrated that the NPs has strong absorbance in second near infrared window and the adsorbed energy can induce the local temperature rise and the acoustic wave for the photothermal therapy and photoacoustic imaging. The ex vivo and in vitro experiments demonstrate that with the modified targeting peptide, the NPs can accumulate more in the thrombosis lesion and under the NIR-II laser, and that the NPs can be used as PA probe and thrombolytic tool. The in vivo experiments further demonstrate the NPs synergistic capacity for the diagnosis and treatment of thrombosis.

Although a very interesting study and well supported by the data, the major issue is the NIR-II strategy is not very novel. Authors should demonstrate in introduction the advantages of their nanosystem compared to several of those already reported in the literature, as well as the potential clinical translation.

Response: We sincerely thank the reviewer for your positive feedback and valuable suggestions to our work.

In this work, we first developed a novel kind of semiconducting homopolymer via rational molecular design that exhibited excellent NIR-II light harvesting ability and violent molecular motion to boost the photothermal conversion and PA signal. Then, a thrombus-targeting theranostic nanoplatform was developed for the NIR-II PA imaging-based sensitive thrombosis diagnosis and robust thrombolytic treatment by combining thrombus-localized photothermal effect and on-demand NO release. Although the NIR-II PA imaging and/or NIR-II photothermal effect have been reported for biomedical applications, most of them focused on cancer detection and therapy. To the best of our knowledge, this is the first example of the development of theranostic nanoplatform for NIR-II PA imaging of thrombus and synergistic thrombolysis therapy. Thus, both the materials and biomedical application in this work are different from previous reports, and we hope that this work will give inspirations for the further development of advanced NIR-II theranostic platform via rational molecular design to maximize their photophysical transformation and performance in novel biomedical applications.

PA imaging that integrates light excitation and ultrasound detection has higher spatial resolution and deeper penetration depth when compared with conventional fluorescence imaging. And the NIR-II light-excited PA imaging displays further improved imaging contrast and penetration depth as the longer-wavelength NIR-II light has significantly reduced light attenuation in biological tissue and minimal background interference. In addition, compared with inorganic materials, the organic polymer-based NIR-II PA contrast agent developed here possessed the merits of relatively good biocompatibility, well-defined structure and excellent reproducibility. These features make the NIR-II theranostic nanoplatform fabricated in this work promising for potential clinical translation. However, the photophysical properties of nanoprobes still need to be carefully optimized, and their long-term safety and in vivo transport and metabolism require a thorough understanding before clinical research. According to the reviewer's comments, we have added some discussions about the novelty and potential clinical translation of our nanosystem in the corresponding section of the revised manuscript (line 11-20, page 5).

Furthermore, other major issues are indicated below.

1. In Figure 3 a, the observed size of the NPs in the TEM is not inline with the DLS results and the claimed 126 nm in the text. This needs to be further clarified.

Response: According to the TEM images, SP NPs had the spherical structure with an average size of around 110 nm. The DLS results indicated that the hydrodynamic diameters of SP NPs were about 126 nm. The diameter for dried NPs measured by TEM was usually smaller than the hydrodynamic diameter determined by DLS, which was likely due to the drying and shrinkage of NPs during the TEM sample preparation. This phenomenon has also been reported by previous papers (Nat. Commun. 2018, 9, 2053; ACS Nano 2017, 11, 7177; Small 2023, 19, 2207995; J. Colloid Interface Sci. 2021, 604, 208). According to the reviewer's suggestion, we have provided both the TEM and DLS results in the revised manuscript and explained the difference between them (line 23-27, page 7).

2. For the NPs fabrication, authors should demonstrate explain how to make sure the encapsulation of BNN6 and SP NPs is uniform through the self-assembly of DSPE-PEG.

Response: The polymer PTIIG and BNN6 were readily assembled into NPs via the nanoprecipitation method with DSPE-PEG₂₀₀₀ as the surfactant. During the nanoprecipitation process, the hydrophobic polymer PTIIG and BNN6 were expected to randomly self-assemble to form the NP core, and the amphiphilic DSPE-PEG would further assemble on NP surface through hydrophobic interactions between the lipophilic alkyl chains and the NP core (Scheme 1). Nanoprecipitation method has been widely used for the fabrication of hydrophobic compound-loaded and DSPE-PEG-coated polymer NPs, and this method has also been demonstrated to have a good reproducibility (Int. J. Pharm. 2017, 532, 66; Molecules 2020, 25, 3731). In addition, we further measured the sizes, polydispersity index (PDI), and compound encapsulation efficiency of NPs prepared from different batches, and the results also indicated that the NPs prepared by nanoprecipitation method here had good size uniformity and reproducibility (Table 1). The results have been added (new Supplementary Fig. 17) and discussed in the revised manuscript (line 19-21, page 9).

Scheme 1. Schematic illustration of the nanoprecipitation process for formulating hydrophobic PTIIG and BNN6 molecules and amphiphilic DSPE-PEG into nanoparticles.

Table 1. The NPs prepared from different batches were analyzed for their hydrodynamic diameters, PDI and the encapsulation efficiencies of PTIIG and BNN6.

Batch	Size (nm)	PDI	Encapsulation efficiency	
			PTIIG	BNN6
1	166.5±3.5	0.19±0.03	59.4%	42.6%
2	168.3±3.5	0.18±0.01	62.1%	40.1%
3	167.8±4.1	0.19±0.01	60.9%	41.5%
4	169.4±4.7	0.2±0.02	62.5%	39.5%
5	167.4±2.8	0.19±0.02	58.9%	40.9%

3. For the photothermal therapy, 1064 nm laser with 1W/cm² was used for 10 minutes' laser irradiation. With this high power, could it be harmful for the local treatment area? Because for the thrombosis laser treatment, the vessel rupture, hemorrhage and the purpura can occur, the tail bleeding experiment cannot show the hemorrhagic side effect of the local treatment area. Further experimental demonstrations should be provided here.

Response: We thank for the reviewer's suggestion. The maximum permissible exposure (MPE) of 1064 nm laser for skin (ANSI Z136.1–2007, American National Standard for Safe Use of Lasers) is 1 W/cm² (J. Am. Chem. Soc. 2014, 136, 15684; Adv. Mater. 2018, 30, 1705980; Angew. Chem. Int. Ed. 2019, 58, 15526), which is much higher than that of short-wavelength laser (e.g., the MPE of 808 nm light is 0.33 W/cm²). So the 1064 nm laser power in this work is within the safety range. In addition, we also further investigated whether the NIR-II light irradiation used in this work

would lead to blood vessel injury. After light exposure, the treated carotid arteries of mice were collected and sectioned for H&E, elastic fiber and reticular fiber staining, and they were also evaluated by terminal deoxynucleotidyl transferase – mediated deoxyuridine triphosphate nick end labeling (TUNEL) assay. The results indicated no obvious damage to the nearby blood vessels (Fig. 1), including the elastic plate deformation and inflammatory reaction after the treatment of B@SP-C NPs + NIR-II light irradiation. The results have been added (new Supplementary Fig. 36) and discussed in the revised manuscript (line 3-9, page 18).

Fig. 1. Cross-sectional histology analysis of normal mice blood vessels and the 1064 nm NIR-II laser (1 W cm^{-2}) irradiation-treated thrombotic vessels (stained with H&E, elastic fiber staining, reticular fiber staining and TUNEL). Scale bars = 100 μm .

4. Authors should provide more TEM pictures of the NPs (SP NPs, B@SP NPs, and B@SP-C NPs) in lower magnification with more particles for the uniformity determination.

Response: According to the reviewer’s nice suggestion, we have provided the TEM images of different NPs in lower magnification, and more NPs were included in these pictures. The new images were shown in Supplementary Fig. 11.

Fig. 2. Representative TEM images of (a) SP NPs, (b) B@SP NPs, and (c) B@SP-C NPs. Scale bars = 500 nm.

5. In the text, the order of Figure S9 and Figure S10 is reversed.

Response: We sincerely thank the reviewer for your careful reading and pointing out this. We have changed the order of Figure S9 and Figure S10, and also carefully checked the whole manuscript to avoid such errors.

6. Authors should provide the FTIR spectrum of free CREKA, otherwise, it cannot be claimed in the text that “the characteristic bands of BNN6 and CREKA peptide appeared in the FTIR spectrum of B@SP-C NPs” since the spectra in the Figure S11 are similar.

Response: According to the reviewer’s suggestion, we have also provided the FTIR spectra of free CREKA, and the results indicated that some of the characteristic bands of CREKA peptide (such as the amide I band of the peptide backbone at 1674 nm, Langmuir 2022, 38, 12905) obviously appeared in the spectrum of B@SP-C NPs. However, just as the reviewer mentioned, these spectra are similar. Thus, we also provided the ¹H NMR spectra of DSPE-PEG-Mal and DSPE-PEG-CREKA to indicate that we have successfully conjugated CREKA with DSPE-PEG-Mal to obtain the DSPE-PEG-CREKA. The disappearance of the chemical shift of maleimide group at 7.0 ppm indicated the successful conjugation of CREKA to DSPE-PEG-Mal (Fig. 3). The results were well supported by previous reports (Bioact. Mater. 2023, 20, 208; Nano Today 2020, 35, 100986). The results have been added to the revised manuscript (new Supplementary Fig. 10,15).

Fig. 3. ^1H NMR spectra of DSPE-PEG-Mal and DSPE-PEG-CREKA (400 MHz, DMSO-d_6).

7. Authors should explain how to evaluate clearance of the free injected UK as comparison, based on blood test or fluorescence.

Response: The clearance of free UK in blood was measured by UK assay kit according to previous papers (Nat. Commun. 2021, 12, 6242; Transl. Stroke Res. 2021, 12, 844). Following the intravenous injection of free UK, the blood was drawn from mice at predetermined time points. The obtained fresh blood was immediately centrifuged to get the supernatant serum, which was then analyzed with the UK assay kit (Urokinase Activity Assay Kit, Abcam) according to the manufacturer's instructions. We have provided more descriptions about the UK measurement in the revised manuscript (line 19-22, page 26).

8. Since the synthesized SP NPs and B@SP NPs are hydrophobic, Authors should provide more information about how to disperse those NPs in the injection solvent.

Response: In this work, the hydrophobic PTIIG and BNN6 could readily assemble into water-soluble NPs via the nanoprecipitation method with the amphiphilic polymer DSPE-PEG₂₀₀₀ as surfactant. Thus, the as-synthesized SP NPs, B@SP NPs, and B@SP-C NPs exhibited good water solubility (Photos below). Nanoprecipitation is a widely used method to formulate NPs, during which the hydrophobic organic molecules randomly assemble in the core, and the amphiphilic surfactant form the shell layer coated on NP surface (Adv. Drug Deliv. Rev. 2014, 71, 86; Nat. Nanotechnol. 2014, 9, 233; Nano Lett. 2017, 17, 606). The general process of nanoprecipitation is

shown in Scheme 1. In our case, PTIIG, BNN6 and the amphiphilic surfactant (DSPE-PEG₂₀₀₀) were dissolved in organic solvent (tetrahydrofuran, THF), which was then dropped into aqueous medium, and the organic solvent was then evaporated under stirring. During this process, the hydrophobic organic molecules were expected to randomly self-assemble in the core, and the amphiphilic DSPE-PEG formed the shell with hydrophilic PEG extended on the surface. We have added more descriptions about the NP preparation process in the revised manuscript (line 17-20, page 7).

Fig. 4. Photographs of the solution of SP NPs, B@SP NPs, and B@SP-C NPs.

Scheme 1. Schematic illustration of the nanoprecipitation process for formulating hydrophobic PTIIG and BNN6 molecules and amphiphilic DSPE-PEG into nanoparticles.

9. Finally, another major issue is the rather low number of animals used for each of the indicated animal experiences.

Response: According to the reviewer's nice suggestion, we have increased the numbers of animals used for the major in vivo NIR-II PA imaging (Fig. 6c,d and Supplementary Fig. 27 in the revised

manuscript) and thrombolysis experiments (Fig. 7c and Fig. 9d in the revised manuscript) (from $n = 3$ to $n = 5$). The new results were consistent with previous data and well supported our conclusion.

Responses to the Comments and Suggestions of Reviewer #2

Reviewer #2 (Remarks to the Author):

The authors show a very nice and concise story for a theranostic imaging of thrombi using PA imaging. Further comments primarily apply to the PA imaging. While the PA study is nicely done and seems convincing to me I would appreciate a number of additions and controls:

1) Other works showing thrombus detection with NPs in PA (for example: 10.2147/IJN.S216603) should be discussed. And comparisons on the performance would be worthwhile.

Response: We sincerely thank the reviewer for your careful reading and positive feedback to our work. According to the reviewer's nice suggestion, we have discussed other related works, and also compared our nanoprobe with others in the revised manuscript (line 11-20, page 5). Although several papers have reported the detection of thrombus using PA imaging (e.g., Int. J. Nanomed. 2019, 14, 7155; ACS Nano 2018, 12, 392), most of them were based on NIR-I PA imaging and the use of commercially available contrast agents. In our work, we first designed and synthesized a novel kind of high-performance NIR-II PA probe, and then demonstrated that the NIR-II PA imaging of thrombus could provide higher signal-to-noise ratio (SNR) and deeper tissue penetration than traditional NIR-I PA imaging as the longer-wavelength NIR-II light has significantly reduced light scattering and attenuation in tissue and minimal background interference. To the best of our knowledge, this is the first example of the development of theranostic nanoplatform for in vivo NIR-II PA imaging of thrombus and synergistic thrombolysis therapy.

2) Why is the different absorption (3f) in no way reflected in the PA signals (4f). Given the difference in absorbance the PA signal should at least be slightly different for the 3 different NPs, no?

Response: We sincerely thank the reviewer for your careful reading. When we measured the absorption spectra of these three NPs, we just conducted the qualitative analysis as we mainly focused on the location of absorption bands, and did not make these NPs at the same concentration of PTIIG, thus their absorbance at NIR region showed some difference. While for the PA

quantitative analysis in Fig. 4f, the contents of PTIIG in these three NPs at each concentration were exactly the same, thus their PA signals were nearly the same. According to the reviewer's comments, we remeasured the absorption spectra of these three NPs at the equal concentration of 6 $\mu\text{g}/\text{mL}$ based on PTIIG polymer. As shown in Fig. 1, the absorption profiles of these NPs in the NIR region now looked the same, which was consistent with the PA signals. The result has been added to the revised manuscript (Fig. 2i in the revised manuscript).

Fig. 1. Absorption spectra of SP NPs, B@SP NPs, and B@SP-C NPs at the same concentration of 6 $\mu\text{g}/\text{mL}$ based on PTIIG.

3) Please add info on how the anatomy BG images in 7a and c are obtained. And how the false color? Is the anatomy always the same WL? If so, why are they looking slightly different in the different WL. If at different WL how is the false color generated – just the raw signal at the indicated WL?

Response: These merged images were obtained by superimposing each PA image in pseudo-color on the gray-scale US image. The pseudo-colored PA images were reconstructed by Vevo LAB 5.6.1 software, and the intensity of the pseudo-color in each PA image reflected the intensity of local PA signals. Fig. 7a (Fig. 6a in the revised manuscript as one reviewer suggested us to remove some figures to Supplementary information) showed the representative PA images of mice carotid artery upon the excitation of 800, 950, 1260 or 1350 nm light, respectively. When the samples were irradiated with different wavelengths of laser, the nanoprobe and some endogenous tissue chromophores would generate different strengths of acoustic signals, which were then converted

to the pseudo-color with different intensities. In Fig 7c (Fig. 6c in the revised manuscript), all the PA images were obtained with the irradiation of 1260 nm laser. As B@SP-C NPs had superior PA imaging and thrombus-targeting abilities, the thrombus sites of the mice injected with B@SP-C NPs showed higher PA signals (brighter red pseudo color) than those of mice injected with non-targeted B@SP NPs or PBS. According to the reviewer's question, we have added more descriptions about the PA images and pseudo-color in the corresponding section of the revised manuscript (line 27-30, page 7; line 4-5, page 39).

4) Signals outside artery not present in PBS, where do they come from? E.g. 7c top row 30min.

Response: In Fig. 7c, the weak signal outside thrombus site might originate from the non-specific distribution of nanoprobes in some surrounding vessels and tissues. In particular, some tiny vessels might be injured during the construction of carotid artery thrombus model and the isolation of carotid vessels, and nanoprobe might slightly bind these sites. However, for the mice injected with the thrombus-targeting B@SP-C NPs, these non-specific signals were much weaker than the PA signals from thrombus site.

5) It would be nice to show multiple slices (in the suppl) of the in vivo PA images to show continuity of the artery PA signal.

Response: We sincerely thank the reviewer for the constructive suggestion. According to the reviewer's suggestion, the thrombotic carotid artery was scanned at the indicated direction to acquire multiple slices of PA images at different depths (Figure below). These images not only indicated the continuity of PA signals but also provided the 3D tomography information about the target site. These PA images have been added in the revised manuscript (new Supplementary Fig. 30).

Fig. 2. The thrombus region of mice injected with B@SP-C NPs was scanned in the Z direction to acquire multiple slices of PA images. Slide step = 0.05 mm, Scale bars = 2 mm.

6) Please show all mice data as similar scaled images in the supplement. The error bars (7d) for a set of in vivo measurements I find impressively small.

Response: In Fig. 7d (new Fig. 6d in revised manuscript), the error bars for the groups of B@SP NPs and PBS seem relatively small, which was due to the following reason: only weak or negligible PA signals were detected from the two groups, thus the deviations were also small, and the large scale of y axis currently used made the error bars seem even smaller. As the reviewer suggested, we have also provided all the PA images of Fig. 7d in the revised Supplementary information (new Supplementary Fig. 27). In the new figures, the results for total five mice were given and quantified as one reviewer suggested us to increase the numbers of animals used.

Fig. 3. The in vivo NIR-II PA/US merged images of the thrombotic artery at different time points as indicated after the mice were treated with B@SP-C NPs, B@SP NPs, or PBS, respectively. Five mice were included in each group, and the representative images of one mouse from each group were shown in Fig. 6c in the main text, and the other four mice results were presented here. The grey and red bars represent ultrasound and PA signals, respectively. Scale bars = 2 mm.

7) With a tunable laser source as you have in the LAZR-X system you could scan around the peak of your agent and even improve detectability by spectral unmixing. It would be nice when this could be added to show the potential of the method.

Response: We thank for the reviewer's great suggestion. We conducted the PA imaging using different excitation light wavelengths (800, 950, 1260, and 1350 nm). Then the PA images at different wavelengths were unmixed using the Vevo LAB 5.6.1 software based on the PA spectral signatures of nanoprobe and other endogenous absorbers. As shown in Fig. 4, the PA signals from the nanoprobe and multiple endogenous absorbers (including oxygenated (HbO₂) and deoxygenated hemoglobin (Hb), lipid, and water) could be separated. Thus, the multi-spectral PA imaging and spectral unmixing technique could help to eliminate background interference and optimize the contrast, allowing the nanoprobe-labeled thrombus site to be more clearly and accurately visualized. According to the reviewer's nice suggestion, we have added the spectral unmixing results in the revised manuscript (new Supplementary Fig. 25).

Fig. 4. Spectral unmixed results for the in vivo NIR-II PA images of thrombus at different wavelengths to separate the signal from B@SP-C NPs and multiple endogenous absorbers. Scale bar = 2 mm.

Responses to the Comments and Suggestions of Reviewer #3

Reviewer #3 (Remarks to the Author):

This manuscript seeks to demonstrate a polymer nanoplatfom that integrates the long-wavelength near infrared-II (NIR-II) photoacoustic (PA) imaging with nanoparticles delivery system. This approach is interesting to address the unmet clinical need for detecting thrombus formation and to provide anti-thrombotic activity in the cardiovascular space. The authors demonstrated a semiconducting homopolymer nanoparticles with fibrin-specific peptide for NIR-II absorbance, photo-to-heart conversion, and selective thrombus targeting. However, the performance of the nanoparticle would be strengthened with data to support selectivity, potency, and biocompatibility. The following are major and minor comments to address the results and conclusion.

Major Comments:

1. Unclear are how to accumulate the nanoparticles near the thrombosis, and how to demonstrate the selectivity of fibrin-specific peptide. Please provide the concentration of nanoparticles near the regions of thrombosis as compared to the non-thrombosed blood vessels.

Response: We sincerely thank the reviewer for your positive feedback and valuable comments to our work.

During thrombus progression, fibrin was generated upon the activation of coagulation cascades, which was deposited both inside and on the surface of the thrombus to stabilize thrombus (Nature 2008, 451, 914; Biomaterials 2014, 35, 2961). As one of the main components of thrombus, fibrin is an important target for site-specific delivery to thrombi. The CREKA pentapeptide is known to display a very high affinity for fibrin, thus endowing B@SP-C NPs with thrombus-targeting ability (Proc. Natl. Acad. Sci. USA 2009, 106, 9815; Proc. Natl. Acad. Sci. USA 2007, 104, 932; Biomaterials 2014, 35, 2961). To demonstrate the thrombus-targeting ability of our NPs, the binding of B@SP-C NPs or the non-targeted B@SP NPs towards both in vitro artificial thrombus clots and in vivo carotid artery and lower extremity arterial thrombus were compared (Fig. 4b, 5c, 9b in the main text). The results indicated that B@SP-C NPs possessed superior thrombus targeting ability than the non-targeted B@SP NPs. In addition, for the control healthy carotid artery, B@SP-C NPs showed little binding affinity (Fig. 5c in the main text). In another control experiment, free CREKA was administrated into mice prior to B@SP-C NPs injection to occupy

the fibrin binding sites, and then the accumulation of B@SP-C NPs in thrombus was found to be dramatically decreased (Fig. 5c in the main text), suggesting that the binding of B@SP-C NPs with thrombus was mediated via the fibrin-targeting CREKA. All these results collectively indicated that the CREKA-modified B@SP-C NPs had good thrombus-targeting ability. In the revised manuscript, we also co-incubated the Dil-loaded B@SP-C NPs with FITC-labeled fibrin, and demonstrated that the red fluorescence from NPs was well co-localized with the green fluorescence from fibrin, suggesting the specific interaction between them. The results have been added (new Supplementary Fig. 16) and discussed in the revised manuscript (line 14-17, page 9).

Fig. 1. The colocalization of Dil-loaded B@SP-C NPs (red fluorescence) with FITC-labeled fibrin (green fluorescence) indicating the binding between them. Dil is 1,1-dioctadecyl-3,3,3',3'-tetramethylindocarbocyanine perchlorate and FITC is fluorescein isothiocyanate. Scale bars = 50 μm .

According to the reviewer's suggestion, we have also measured the concentrations of B@SP-C NPs in the region of thrombosis. The DiR-loaded B@SP-C NPs were injected into the carotid thrombosis-bearing mice, and then both the thrombotic carotid artery and the contralateral healthy carotid artery were collected 1 hour post NP administration. The vessel tissues were homogenized and DiR in tissue homogenate was extracted by methanol, and a standard curve between the fluorescence intensity and DiR concentration was built to calculate the concentration of DiR in blood vessels. As a result, the concentration of B@SP-C NPs in the thrombosis site was evaluated to be about 400 ng/mg, which was much higher than that in normal blood vessels (< 20 ng/mg). We have added the results (new Supplementary Fig. 24) and discussion in the revised manuscript (line 23-25, page 13).

Fig. 2. (a) The relationship of fluorescence intensity and different concentrations of DiR in methanol. (b) The concentrations of B@SP-C NPs in normal healthy vessel and thrombosis vessel. Data are presented as mean \pm s.d. ($n = 3$).

2. Please address how the nanoparticles are degraded and secreted from the circulation.

Response: We thank for the reviewer's suggestion. To study the metabolism process, the feces and urine of mice injected with the DiR-loaded B@SP-C NPs were collected at different time points post NP administration. As shown in Fig. 3, obvious fluorescent signals were observed in the collected feces rather than urine on day 1 and day 3, and little NP fluorescent signals could be detected in feces on day 7, suggesting that B@SP-C NPs could be almost completely excreted from the mouse body through biliary pathway after 7 days. The results were consistent with previous reports (ACS Nano 2017, 11, 7177; Adv. Funct. Mater. 2023, 33, 2212380; Proc. Natl. Acad. Sci. USA 2008, 105, 1410). The results have been added (new Supplementary Fig. 23) and discussed in the revised manuscript (line 30, page 12; line 1-2, page 13).

Fig. 3. Representative fluorescent images of the collected feces and urine at various time points after intravenous injection of DiR-loaded B@SP-C NPs into mice.

3. Please provide the duration required to remove the thrombus formation and the percentage of thrombus that can be removed using the on-demand NO release. Using NO to remove thrombosis is recognized to require a long duration, whereas the half-life of NO is short. Please demonstrate the thrombus removal efficiency by NO in response to FeCl₃-induced carotid thrombosis and lower extremity arterial thrombosis models.

Response: Thank the reviewer for the kind suggestion. As the on-demand NO release was activated by the PTT effect, we could not establish the single NO treatment group. Thus, to determine the thrombus removal efficiency mediated by the on-demand NO release, the therapeutic outcome of “SP NPs + NIR-II” and “B@SP NPs + NIR-II” groups were compared, and the main difference between the two groups was whether NO release existed or not. As shown in Fig. 7c and 8d in the main text, the single photothermal thrombolysis only displayed moderate effect, and the blood vessels were rapidly blocked again after the treatment. While, with the aid of controlled NO generation, a more efficient thrombus removal was achieved. When compared to “SP NPs + NIR-II” group, the blood perfusion in the “B@SP NPs + NIR-II” group increased from 30% to 68% in carotid thrombosis model and increased from 23% to 63% in lower extremity arterial thrombosis models at 2 h post treatment. As reported previously, NO was able to reduce thrombus formation by regulating vasodilation and inhibiting platelet adhesion and aggregation (Nat. Rev. Cancer 2009, 9, 182; Chem. Soc. Rev. 2012, 41, 3742; J. Thromb. Haemost. 2003, 1, 2112.) and the controlled delivery system used in this work could greatly increase the duration of NO in thrombotic lesion. In addition, the in-situ NO gas generation might also promote the mechanical thrombolysis and the deep penetration of NPs into thrombus to boost antithrombotic outcomes (Nat. Commun. 2019, 10, 966). We have provided more explanations about the NO-mediated thrombolysis in the revised manuscript (line 28-31, page 15).

5. In Figure 3, dynamic light scattering, and transmission electron microscope should have different results to measure the average diameter of nanoparticles.

Response: Yes, we totally agree with the reviewer. According to the TEM images, SP NPs had the spherical structure with an average size around 110 nm. The DLS results indicated that the hydrodynamic diameters of SP NPs were about 126 nm. The diameter for dried NPs measured by TEM was usually smaller than the hydrodynamic diameter determined by DLS, which was likely due to the drying and shrinkage of NPs during the TEM sample preparation. This phenomenon has also been reported by previous papers (Nat. Commun. 2018, 9, 2053; ACS Nano 2017, 11, 7177; Small 2023, 19, 2207995; J. Colloid Interface Sci. 2021, 604, 208). According to reviewer's suggestion, we have provided both the TEM and DLS results in the revised manuscript and explained the difference between them (line 23-27, page 7).

6. In Figure 4b, please follow a standard protocol to measure PA signal of B@SP-C NPs and provide the references. Figure 4b shows plateaus after 10 mm penetration depth, whereas light attenuation at different penetration depths is exponential. The plateaus could suggest the presence of stray light or other sources of illumination that bypassed the thickness. Thus, the authors should double-check the experiment settings and investigate the sources of the observed plateau. In Figure 4h, how long could B@SP-C NPs endure under laser irradiation? Please enlarge the time of laser irradiation.

Response: According to the reviewer's suggestion, we have provided the reference (Adv. Mater. 2020, 32, 2001146) that we referred for the measurement of in vitro PA signal of NPs at different tissue depths. Under the 800 or 950 nm laser irradiation, the PA signal decreased slowly after 10 mm of penetration depth, the possible reason is as follow: the PA signals at such penetration depth were already very weak, thus the change of PA signal became slow. To observe the complete signal attenuation, we further increased the depths of chicken tissue to 60 mm. As shown in Fig. 4, the decrease of PA intensity with the depth of tissue basically exhibited approximate exponential relationship, and the PA signal of NPs irradiated with 1260 nm light showed the best tissue penetration depth.

For the photostability experiments, according to the reviewer's suggestion, we have prolonged the time of laser irradiation to 60 min, and the result is presented in Fig. 5. During the 60 min of laser irradiation, nearly no decrease of absorption intensity was observed for B@SP-C NPs, while the NIR absorption of ICG decreased to approximate zero in 20 min. This result further suggested the

excellent photo-stability of the PA probe we synthesized. The new results have been added to the revised manuscript (Fig. 3c,h in the revised manuscript).

Fig. 4. PA intensities of B@SP-C NPs ($100 \mu\text{g mL}^{-1}$ based on PTIIG) under various thicknesses of chicken breast upon 800, 950, 1260, or 1350 nm laser irradiation.

Fig. 5. Plots of A/A_0 of B@SP-C NPs and ICG under laser irradiation for different time. A and A_0 are the absorption intensity of B@SP-C NPs (1260 nm) or ICG (780 nm) after and before continuous laser irradiation, respectively. Inset shows the photographs of (i, ii) B@SP-C NPs and (iii, iv) ICG solution before and after light irradiation.

7. In Fig. 6a, the B@SP-C NPs exhibited an improved half-life. However, the graph also indicates that after 25 hours, approximately 30% of the injection remained unclear. Therefore, it is essential to evaluate whether the residual of the NPs after a prolonged period could pose a concern.

Response: Thanks for the reviewer's suggestion. First, we have prolonged the time period for blood circulation profile measurement. As shown in Fig. 6, nearly all the B@SP-C NPs were cleared from blood at 48 h after intravenous injection. One week after the NP treatment, the main organs and blood of mice were collected for biosafety evaluation. The H&E staining of major organs did not show any noticeable pathological changes or inflammatory response post NPs treatment (Supplementary Fig. 33). Moreover, the hematology parameters and multiple hepatorenal indicators of mice were all within the normal ranges (Supplementary Fig. 34,35). These data collectively suggested the good in vivo biocompatibility of B@SP-C NPs. In further, we will conduct long-term and more thorough biosafety assessment.

Fig. 6. Blood circulation profile of B@SP-C NPs and UK after intravenous injection into mice. Inset shows the fluorescence images of blood drawn from mice at the indicated time points after B@SP-C NPs injection. Data are presented as mean \pm s.d. ($n = 3$).

8. The legend of Fig. 7 indicates that the images shown are NIR-II PA images. However, it is unclear whether these are purely PA images or a combination of PA and ultrasound images. The significance of the two different color bars used was not explained.

Response: We are sorry for the unclear description. These images are merged images that obtained by superimposing each PA image in pseudo-color on the gray-scale US image. We have made clear explanation about the merged images and two different color bars used in the revised figure and figure legend (line 5-6,10-11, page 39).

9. The selected field of view in some of the images was too small for adequate comparison. Therefore, the authors may consider including larger or closer images of the region of interest to enable better comparison.

Response: According to the reviewer's suggestion, we have provided the larger PA images of Fig. 6c in the revised Supplementary information (new Supplementary Fig. 26) to enable better comparison.

10. In Fig. 7a, the 800 nm and 950 nm wavelengths showed very strong surface signals, whereas the 1260 nm and 1350 nm wavelengths did not show any. The cause of the strong surface signals should be explained. Moreover, comparing SNR across these wavelengths can be challenging due to differences in water absorption, and the authors may need to confirm that the same laser influence after water is present across all wavelengths.

Response: We thank for the reviewer's nice suggestion. The PA images at different wavelengths were unmixed using the Vevo LAB 5.6.1 software based on the PA spectral signatures of nanoprobe and other endogenous absorbers to figure out the origin of PA signal. As shown in Fig. 7, the PA signals from the nanoprobe and several endogenous absorbers (including oxygenated hemoglobin (HbO₂), deoxyhemoglobin (Hb), lipid, and water) could be separated. The results indicated that the strong background signals in the PA images under 800 and 950 nm light excitation were mainly from HbO₂ and Hb. While the influence of HbO₂ and Hb on the long-wavelength PA images could be negligible due to their weak absorbance in NIR-II window (Nat. Biotechnol. 2006, 24, 848; Chem. Rev. 2015, 115, 10816). Under our experimental conditions, the water signal from tissue could hardly be detected at all the measured wavelengths. Although the absorption of water in NIR-II region is stronger than that in visible and NIR-I spectral region, its absorption peak was located at ≈ 1450 nm and its absorption coefficient in the spectral region below

1350 nm is still very low ($< 1 \text{ cm}^{-1}$). Therefore, the influence of water on our thrombus PA imaging at 1260 nm is weak. We have added these results in the revised Supplementary information (new Supplementary Fig. 25) and discussed them in the revised manuscript (line 11-18, page 14).

Fig. 7. Spectral unmixing of the in vivo NIR-II PA images of thrombus site with the excitation of 800, 950, 1260 and 1350 nm laser after different treatments. Scale bar = 2 mm.

11. In Fig. 8c, the authors should clarify how the blood perfusion was quantified for the entire image or a selected field of view. It is unclear whether some of the figures demonstrate an increase in flow, whereas the corresponding plot shows otherwise. Finally, the color bar used only shows high vs. low. Please provide a quantitative scale that correlates to specific flow numbers.

Response: Thanks for the reviewer's nice suggestion. The blood perfusion was quantified based on a selected field of view. Since the carotid thrombosis model was built in the right carotid artery of mice, the right carotid artery with thrombosis was selected as the region of interest to monitor the blood perfusion after various treatments. The blood flow was recorded by the laser speckle blood flow monitoring system (RWD Life Science) and quantified by LSCI V1.0.0 software. In the revised manuscript, to make the results clear for readers to understand, we have marked the selected field of view for blood perfusion quantification with dotted white rectangle, and also provided the quantitative scale bar that correlates to flow numbers (Fig. 7c,9d in the revised manuscript).

Minor issues:

1. In the third paragraph of the introduction, please provide the penetration depth ranges of different optical technology in the human body.

Response: According to previous reports, the penetration depths of NIR-I and NIR-II fluorescence imaging in tissue are ~3 mm and 1-2 cm, respectively (Adv. Mater. 2018, 30, 1802394; Adv. Funct. Mater. 2019, 29, 1901480). For PA imaging, the tissue penetration depths of NIR-I and NIR-II PA imaging are 2-3 cm and 5-6 cm, respectively (Nat. Commun. 2018, 9, 2898; Nano Lett. 2017, 17, 4964). We have provided the information in the introduction part of the revised manuscript (line 23-25, page 3).

2. In the Introduction, the authors stated that NIR-II light-excited PA imaging could provide higher resolution and signal-to-noise ratio (SNR), as well as deeper tissue penetration due to reduced light scattering and attenuation. However, the resolution of photoacoustic computed tomography is not dependent on the wavelength used. Additionally, the NIR-II light is known to be influenced by water and lipid absorption, which may not improve the SNR as compared to NIR-I.

Response: We sincerely thank the reviewer for pointing out this. Yes, there are several endogenous absorbers, such as oxyhemoglobin (HbO₂), deoxyhemoglobin (Hb), lipid, and water that may influence the PA imaging performance (J. Biomed. Opt. 2019, 24, 040901; Chem. Soc. Rev. 2018, 47, 4258; Nat. Biomed. Eng. 2017, 1, 0010; Photoacoustics 2014, 2, 12). As shown in Fig. 8, HbO₂ and Hb show relatively low absorption in the NIR-II window. Despite water exhibits an absorption peak at ~1450 nm, its absorption coefficient in the spectral region below 1350 nm is still very low. According to these facts and the reviewer's comments, we have changed "NIR-II window" to "NIR-II window of 1000–1350 nm" in the revised introduction.

In addition to the reduced light absorption by tissue in NIR-II window at 1000 to 1350 nm, the scattering coefficients of various tissues toward light are also found to decrease with an increase of wavelength (Fig. 8). The reduced light absorption and scattering by biological tissue in NIR-II window at 1000 to 1350 nm would improve the signal-to-noise ratio and penetration depths of PA imaging. These statements have also been supported by previous papers (J. Biomed. Opt. 2019, 24, 040901; Chem. Soc. Rev. 2018, 47, 4258; Nat. Biomed. Eng. 2017, 1, 0010; Adv. Mater. 2018,

30, 1802394). In the revised introduction, according to the reviewer's comments, we have made some changes to make the description about "NIR-II PA imaging" more clear and rigorous (line 18-20,23, page 3).

Fig. 8. (a) Absorption coefficient spectra of endogenous tissue chromophores at their typical concentrations in the human body. (b) Reduced light scattering coefficients of different biological tissues as a function of wavelength in the 400 to 1700 nm region. These figures were cited from J. Biomed. Opt. 2019, 24, 040901 and Nat. Biomed. Eng. 2017, 1, 0010.

3. Please provide biomedical references to support the statement that “there is growing evidence showing that localized hyperthermia could accelerate the ablation of blood clot via loosening the non-covalent interactions of fibrins, providing a safe and non-invasive thrombolytic approach”. The current references were mostly in chemistry and materials science.

Response: According to the reviewer's suggestion, we have provided biomedical references to support the statement that “localized hyperthermia could accelerate the ablation of blood clot via loosening the non-covalent interactions of fibrins, providing a safe and non-invasive thrombolytic approach” in revised manuscript.

4. Please provide the specific temperature at which the nanoparticles were synthesized.

Response: The nanoparticles were synthesized at the temperature of 20 °C. We have provided the temperature for nanoparticle synthesis in the revised manuscript (line 24-25, page 20).

5. Line 112: There seems to be a typo: "Photo-to-heart" vs. "Photo-to-heat".

Response: We are sorry for the typo. We have corrected it, and also carefully checked the whole manuscript to avoid such typo issues.

6. Fig.1: Please elaborate in the caption of Fig.1 to help understand the schematics.

Response: We thank for the reviewer's nice suggestion. We have given more descriptions about our design in the caption of Fig. 1 in the revised manuscript (line 3-10, page 34).

7. Please provide a scale bar on the fluorescence intensity in Fig. 6e.

Response: Yes, we have provided the scale bar on the fluorescence intensity in Fig. 6e (Fig. 5e in the revised manuscript) in the revised manuscript.

8. Figures 5d and 6e lack the scale bars.

Response: Yes, we have added the scale bars in Fig. 5d and 6e (Fig. 4d and 5e in the revised manuscript) in the revised manuscript.

9. The color bars in the figure are not clearly illustrated. For example, in figures 5b, 6c, 8c, please quantify "high" vs. "low" in the figures or legends.

Response: According to the reviewer's nice suggestion, we have provided the quantitative scale for the color bars in Fig. 5b, 6c, 8c (Fig. 4b,5c,7c in the revised manuscript) in the revised manuscript.

10. This manuscript contains 10 Figures. Some Figures may be considered as supplementary information, such as Figure 2. Please address the units in the synthetic PTIG (Figure 2)?

Response: According to the reviewer's suggestion, some sub-figures in Fig. 2 have been removed to Supplementary information, and we have combined Fig. 2 and Fig. 3 as the new Fig. 2 in the revised manuscript. So the manuscript contains nine figures now. In addition, according to the gel permeation chromatography (GPC) results, the synthetic PTIIG polymer contained about 38 of TIIG repeat unit.

REVIEWER COMMENTS

Reviewer #1 (Remarks to the Author):

The authors have revised the paper extensively, added new experimental work, which strengthen the paper and support the conclusions of the work.
Therefore, my recommendation is to publish in the current form.

Reviewer #2 (Remarks to the Author):

My questions have been sufficiently answered.

Reviewer #4 (Remarks to the Author):

I would like to praise the authors for their very careful and detailed responses to the reviewers' previous comments. They have clearly presented their work on B@SP-C NPs, explaining its integration with NIR-II PA imaging for thrombosis detection and antithrombotic activity. However, I hope they can enhance the paper's quality by addressing the following questions and suggestions.

Some remaining major concerns:

- 1.The author replied to the previous question about strange signal distribution in PACT images. However, For the spectral unmixing via the commercial product, Vevo LAB 5.6.1 software, it is very necessary for the author to understand its principles and limitations and disclose them in the manuscript. The spectrally unmixed images are clear evidence of inaccurate measurements, which showed Hb (blue) mostly on the surface and HbO₂ (green) in scattered areas. In addition, the authors mentioned that the potent surface signal in Fig. 6a at 800 nm and 950 nm could be blood-related, whereas signals at 1260 nm and 1350 nm wavelengths have diminished blood signals. However, the image suggests a stronger blood signal at 950 nm than B@SP-C NPs signals at 1260 nm. To solidify its position as a contrast agent for PA imaging, it could be informative to compare the signal intensities of B@SP-C NPs, water, and blood (Hb and HbO) across various wavelengths.
- 2.In the revised Fig. 7c and Fig. 9d, the reviewer appreciates the author's effort to better show the selected field of view for blood perfusion quantification. With these selected fields of view, there appears to be a significant amount of blood outside the carotid artery area in the SP NPs and SP NPs+NIR II groups, differing greatly from other groups (Fig. 7). Are those indications of hemorrhage after the injection? Will these affect the quantification?

Additional suggestions:

- 1.In Fig. 3g, the authors demonstrated that the PA amplitude remains stable throughout 15 minutes of irradiation. However, Fig. 2 depicts a substantial temperature increase within just 10 minutes of laser irradiation. Intuitively, a temperature change should influence the PA signal rather than it remaining stable since the material Grüneisen parameter depends on temperature. Could the authors please clarify this point?
- 2.There seems to be a discrepancy between the caption of Fig. 3h (showing 780 nm for ICG) and supplementary Fig. 21 (showing an 808 nm laser). Please double-check it.
- 3.I recommend including the molar absorption coefficient (cm⁻¹) on the y-axis of Figure 2i, Supplementary Figure 3, and Supplementary Figure 5. This would help readers better understand its absorption properties as a PA contrast agent.

Responses to reviewers' comments for the manuscript titled "Near-infrared-II photoacoustic imaging and photo-triggered synergistic treatment of thrombosis via fibrin-specific homopolymer nanoparticles".

We sincerely appreciate the reviewers for their positive feedback and valuable comments regarding our article. The point-by-point responses to those comments are outlined below and all modifications made to the manuscript are highlighted by using red colored text. We hope that these revisions meet the reviewers' expectations and the manuscript will be accepted for publication in Nature Communications.

Responses to the Comments and Suggestions of Reviewer #1

Reviewer #1 (Remarks to the Author):

The authors have revised the paper extensively, added new experimental work, which strengthen the paper and support the conclusions of the work.

Therefore, my recommendation is to publish in the current form.

Response: We sincerely thank the reviewer for your positive feedback and support for the publication of our study.

Responses to the Comments and Suggestions of Reviewer #2

Reviewer #2 (Remarks to the Author):

My questions have been sufficiently answered.

Response: We sincerely thank the reviewer for your careful reading and support for the publication of our work.

Responses to the Comments and Suggestions of Reviewer #4

Reviewer #4 (Remarks to the Author):

I would like to praise the authors for their very careful and detailed responses to the reviewers' previous comments. They have clearly presented their work on B@SP-C NPs, explaining its integration with NIR-II PA imaging for thrombosis detection and antithrombotic activity. However, I hope they can enhance the paper's quality by addressing the following questions and suggestions.

Some remaining major concerns:

1. The author replied to the previous question about strange signal distribution in PACT images. However, for the spectral unmixing via the commercial product, Vevo LAB 5.6.1 software, it is very necessary for the author to understand its principles and limitations and disclose them in the manuscript. The spectrally unmixed images are clear evidence of inaccurate measurements, which showed Hb (blue) mostly on the surface and HbO₂ (green) in scattered areas. In addition, the authors mentioned that the potent surface signal in Fig. 6a at 800 nm and 950 nm could be blood-related, whereas signals at 1260 nm and 1350 nm wavelengths have diminished blood signals. However, the image suggests a stronger blood signal at 950 nm than B@SP-C NPs signals at 1260 nm. To solidify its position as a contrast agent for PA imaging, it could be informative to compare the signal intensities of B@SP-C NPs, water, and blood (Hb and HbO) across various wavelengths.

Response: We sincerely thank the reviewer for your positive feedback and valuable suggestions regarding our work.

In this work, in response to a previous reviewer's suggestion to analyze the source of background signals at different imaging wavelengths through spectral unmixing, we conducted spectral unmixing using the commercial software (Vevo LAB 5.6.1) provided by the Vevo® LAZR-X imaging system (Fuji VisualSonics, Canada). The spectral unmixing is based on the PA spectral signatures of each absorber within the sample, and the spectral unmixing algorithm used by this software has been reported in the previous literature (*Photoacoustics* 2013, 1, 36). Currently, several spectral unmixing methods and algorithms have been developed to identify different constituents (*Biomed. Eng. Lett.* 2022, 12, 155; *Nat. Commun.* 2021, 12, 5410). However, the accuracy of spectral unmixing is still affected by the set of wavelengths selected, the overlap of spectral features among different absorbers, local fluence variations, and the high complexity of

biological tissues. To minimize these influences, it is critical to develop exogenous PA contrast agents that exhibit minimal absorption overlap with the sensitive regions of endogenous contrast. The nanoprobe fabricated in this work displayed strong absorption in the range of 1000-1300 nm, where the main endogenous absorbers (such as oxy- and deoxyhemoglobin, lipid, and water) show relatively low absorption, facilitating good spectral differentiation from endogenous contrasts.

From the previous merged spectral unmixed images, it appeared that the distribution of Hb (blue) was mostly on the surface and HbO₂ (green) was in scattered areas. Actually, there were also obvious HbO₂ (green) signals on the surface, which were masked by the blue signal from Hb in the merged images. As shown in the de-merged images (Fig. 1 below and new Supplementary Fig. 25 in the revised Supplementary information), there was a good correlation between the PA signals from Hb and HbO₂, and they both showed stronger signals on the tissue surface compared to deeper tissue regions. This trend might be partly attributed to that the Hb and HbO₂ on the surface could be easily irradiated by laser, and similar results have also been reported by previous literatures (*ACS Nano* 2019, 13, 284; *Photoacoustics* 2020, 18, 100166). To further demonstrate the validity of the spectral unmixing method used here, the solutions of B@SP-C NPs, Hb, HbO₂, lipid, and water were placed at different locations within phantom. And their PA signals over multiple wavelengths were then recorded and the spectral unmixing was conducted using the Vevo LAB 5.6.1 software. As illustrated in Fig. 2 below, the spectral unmixing could accurately differentiate the PA signals from various agents.

Furthermore, as the reviewer suggested, we compared the absorption coefficients and in vivo PA signal intensities of B@SP-C NPs, water, Hb, and HbO₂ at various wavelengths (depicted in Fig. 3 below). Broadly speaking, Hb and HbO₂ showed relatively strong absorption in NIR-I region, which matched well with their bright PA intensity in NIR-I region. On the contrary, B@SP-C NPs manifested strong absorption and thus high PA signal in NIR-II region. Water showed weak PA signals at all the tested wavelengths due to its low absorption coefficient. These results indicated that the PA signal intensity of each absorber at different wavelengths was well correlated with their absorption capabilities in the corresponding region. However, it is noted that the abundance of each absorber within tissue also greatly affects their PA signal intensity. Given the high concentration of Hb and HbO₂ in blood (higher than 100 g/L, *JAMA* 1998, 279, 199; *Lancet* 2002, 359, 1747), a stronger blood PA signal at 950 nm was observed compared to the signal from B@SP-C NPs, even though the absorption capacity of Hb and HbO₂ at 950 nm is lower than that

of our NPs. Nevertheless, it's noteworthy that this has little influence on the in vivo PA imaging performance of our study, as 1260 nm excitation was eventually chosen for in vivo PA imaging, in which the background signal was negligible. These results also highlighted the advantage and necessity of developing highly efficient NIR-II PA probe for thrombus imaging.

In the current study, we mainly focused on the synthesis of a novel kind of high-performance NIR-II PA probe and its integration with on-demand NO generation to fabricate the targeted theranostic nanoplatform tailored for both thrombosis diagnosis and synergistic antithrombotic therapy. In further work, we will try to optimize the spectral unmixing methods to achieve better spectral unmixing analysis outcomes. According to the reviewer's insightful suggestion, we have added some discussion about the principles and limitations of the current spectral unmixing method used in the revised manuscript (line 14-22, page 28). Additionally, we have also included a detailed comparison regarding the absorption coefficients and PA signal intensities of different contrasts at various wavelengths (new Supplementary Fig. 26) as well as the related explanations to offer readers an improved grasp of the PA imaging performance of the nanoprobe (line 20-31, page 14; line 1-2, page 15).

Fig. 1. The spectral unmixing images of the in vivo PA imaging of thrombus site with the excitation of 800, 950, 1260 or 1350 nm laser after injecting different formulations, i.e., (a) PBS, (b) B@SP NPs, and (c) B@SP-C NPs. For the partial overlap of the PA signals from HbO₂ and Hb, the corresponding unmixed results for HbO₂ and Hb were also provided separately. Scale bars = 2 mm.

Fig. 2. The PA signals of the solutions of B@SP-C NPs, Hb, HbO₂, lipid, and water at different wavelengths (with the positions of each solution indicated at the top), and the corresponding spectrally unmixed images (on the right).

Fig. 3. (a) The absorption coefficient and (b) in vivo PA intensity of different contrasts at various wavelengths. The excitation wavelength of 1260 nm (indicated by dotted squares) was finally chosen for in vivo PA imaging.

2. In the revised Fig. 7c and Fig. 9d, the reviewer appreciates the author's effort to better show the selected field of view for blood perfusion quantification. With these selected fields of view, there appears to be a significant amount of blood outside the carotid artery area in the SP NPs and SP NPs+NIR II groups, differing greatly from other groups (Fig. 7). Are those indications of hemorrhage after the injection? Will these affect the quantification?

Response: We greatly appreciate the reviewer's meticulous review. The signals outside the carotid arteries in the laser speckle blood flow monitoring system (LSBFMS) images stemmed from the blood perfusion of the surrounding capillaries, rather than any instances of bleeding. For each mouse, we focused on the carotid artery and monitored the changes of blood perfusion within it. However, for the non-focused surrounding tissues, they might exhibit different blood perfusion signals owing to the variations of individual mice in the positioning, exposure area, and imaging working distance of surrounding tissues. As shown in the bright-field images (Fig. 4 below), there was no obvious bleeding around the carotid artery area for both SP NPs and SP NPs + NIR-II groups, and it also suggested that the positioning of the surrounding tissues for each mouse during the measurement could vary, thereby resulting in different background signal. To indicate the randomness of the surrounding background signal, we present the LSBFMS images from other mice in the SP NPs and SP NPs + NIR-II groups (Fig. 5 and 6 below), which displayed individual variation in surrounding background signals.

As we focused on the carotid artery region of each mouse during LSBFMS imaging and the blood perfusion signal in carotid artery was accurate, thus the variations in non-focused surrounding background signals did not affect the quantitative analysis of the blood vessels in carotid artery. In the revised manuscript, we have provided more explanations about the origins of surrounding background signals in LSBFMS images to ensure ease of interpretation of the results (line 11-15, page 41).

Fig. 4. Bright-field images showing the field of view for LSBFMS imaging. The carotid arteries of mice (indicated with the dotted white lines) in (a) SP NP group and (b) SP NPs + NIR-II group were focused for LSBFMS imaging before and after FeCl₃ induction and different time post treatment. Scale bars = 2 mm.

Fig. 5. The LSBFMS images of the carotid artery area from other mice in the SP NP group before and after FeCl_3 induction and the therapeutic treatments. The dotted white lines indicated the right carotid thrombotic artery, which was selected as the region of interest to quantify the blood perfusion. Scale bars = 2 mm.

Fig. 6. The LSBFMS images of the carotid artery area from other mice in the SP NPs + NIR-II laser group before and after FeCl_3 induction and the therapeutic treatments. The dotted white lines indicated the right carotid thrombotic artery, which was selected as the region of interest to quantify the blood perfusion. Scale bars = 2 mm.

Additional suggestions:

1. In Fig. 3g, the authors demonstrated that the PA amplitude remains stable throughout 15 minutes of irradiation. However, Fig. 2 depicts a substantial temperature increase within just 10 minutes of laser irradiation. Intuitively, a temperature change should influence the PA signal rather than it remaining stable since the material Grüneisen parameter depends on temperature. Could the authors please clarify this point?

Response: We understand the reviewer's concern regarding the potential impact of temperature changes on the PA signal as the material Grüneisen parameter depends on temperature. In this study, the laser source and power intensities used for PA imaging and photothermal measurement were markedly different, so they would cause different temperature changes.

For the photothermal test depicted in Fig. 2 of the manuscript, a high-power laser (1 W/cm^2) with continuous output was employed for irradiation, resulting in a substantial temperature increase. In contrast, for the PA measurements, a pulsed laser with a duration of 7 ns and output power of 10 mJ/cm^2 at a repetition rate of 20 Hz was utilized as the excitation light. Such short-duration and long-break excitation light could hardly induce a noticeable increase in the system temperature. In addition, as it is necessary to have medium to transmit the acoustic waves, both the transducer and the test specimen were immersed in water during the in vitro PA measurement process, which further helped to maintain a stable temperature. Consistent with our findings, some previously reported excellent NIR-I polymer PA agents also showed little PA signal changes after pulsed laser scanning over a duration of time (*Nat. Nanotechnol.* 2014, 9, 233). According to the reviewer's comments, we have included more explanations about the PA measurement procedure and the potential impact of temperature on PA measurement in the revised manuscript (line 1-5, page 26).

2. There seems to be a discrepancy between the caption of Fig. 3h (showing 780 nm for ICG) and supplementary Fig. 21 (showing an 808 nm laser). Please double-check it.

Response: We are grateful for the reviewer's thorough examination. Supplementary Fig. 21 presents the alterations in the entire absorption spectra of ICG before and after 808 nm laser irradiation for different time. According to Supplementary Fig. 21, the absorption intensity of ICG at the wavelength of 780 nm (the peak absorption) after exposure to 808 nm laser irradiation over different durations was plotted, and the resulting graph is depicted as Fig. 3h. While ICG exhibits an absorption peak at around 780 nm, it is usually irradiated by the commercially available 808 nm laser to induce PTT/PA effect, which aligns with previous studies (*Nat. Commun.* 2016, 7, 13193; *Nat. Commun.* 2022, 13, 2794). In response to the reviewer's feedback, we have augmented the figure caption to provide more explanations about the two wavelengths to facilitate readers' comprehension (line 26-28, page 37).

3. I recommend including the molar absorption coefficient (cm-1) on the y-axis of Figure 2i, Supplementary Figure 3, and Supplementary Figure 5. This would help readers better understand its absorption properties as a PA contrast agent.

Response: We greatly appreciate the reviewer's nice suggestion. And we have provided the molar absorption coefficient on the y-axis of Fig. 2i, Supplementary Fig. 3, and Supplementary Fig. 5 in the revised manuscript.

REVIEWERS' COMMENTS

Reviewer #4 (Remarks to the Author):

I am pleased with the improvements made. The paper is well-written, and the research is solid. I recommend that the paper be published.

However, I do have a minor suggestion for the authors to consider before final publication. Specifically, it would be beneficial to further confirm the values of the hemoglobin absorption coefficient mentioned in the revised paper. There are plenty of papers describing the molar absorption coefficient of hemoglobin, which seems quite different than the value in this paper. In addition, the unit of absorption coefficient in Supplementary Fig. 5 (g-1 L-1) is confusing. Please double-check.

We sincerely thank the reviewer for his/her precious time and recognition of our work. Below are our point-by-point responses to the reviewer's comments.

Responses to the Comments and Suggestions of Reviewer #4

Reviewer #4 (Remarks to the Author):

I am pleased with the improvements made. The paper is well-written, and the research is solid. I recommend that the paper be published.

However, I do have a minor suggestion for the authors to consider before final publication. Specifically, it would be beneficial to further confirm the values of the hemoglobin absorption coefficient mentioned in the revised paper. There are plenty of papers describing the molar absorption coefficient of hemoglobin, which seems quite different than the value in this paper. In addition, the unit of absorption coefficient in Supplementary Fig. 5 (g-1 L-1) is confusing. Please double-check.

Response: We sincerely thank the reviewer for your positive feedback and supporting publication of our study.

For the absorption coefficient of hemoglobin, different values have been reported when using different units (Nat. Methods 2016, 13, 639; Sci. Transl. Med. 2019, 11, eaav2169). For example, both “absorption coefficient (cm^{-1})” and “molar absorption coefficient ($\text{cm}^{-1} \text{M}^{-1}$)” are commonly used units, and they yield distinct values. In our paper, when comparing the absorption capabilities of hemoglobin, water, and the polymer imaging agent, we chose to use absorption coefficient as the unit instead of using molar absorption coefficient, given it is hard to define the exact molar concentration of polymer imaging agent. The values of absorption coefficients of hemoglobin presented in this work (Supplementary Fig. 26) align well with those reported in the literatures (Nat. Methods 2016, 13, 639; Biomed. Opt. Express 2011, 2, 600), and we have cited the relevant references in the revised Supplementary Information.

For the unit of absorption coefficient in Supplementary Fig. 5, it should be “ $\text{L g}^{-1} \text{cm}^{-1}$ ”. We are sorry for the typo issue, and we have corrected them in the revised manuscript.